# Multiple stages of evolutionary change in anthrax toxin receptor expression in humans

Lauren A. Choate ⓘ [1,2], Gilad Barshad ⓘ [1], Pierce W. McMahon[1], Iskander Said[2], Edward J. Rice[1], Paul R. Munn[1], James J. Lewis ⓘ [1✉] & Charles G. Danko ⓘ [1,3✉]

The advent of animal husbandry and hunting increased human exposure to zoonotic pathogens. To understand how a zoonotic disease may have influenced human evolution, we study changes in human expression of anthrax toxin receptor 2 (*ANTXR2*), which encodes a cell surface protein necessary for *Bacillus anthracis* virulence toxins to cause anthrax disease. In immune cells, *ANTXR2* is 8-fold down-regulated in all available human samples compared to non-human primates, indicating regulatory changes early in the evolution of modern humans. We also observe multiple genetic signatures consistent with recent positive selection driving a European-specific decrease in *ANTXR2* expression in multiple tissues affected by anthrax toxins. Our observations fit a model in which humans adapted to anthrax disease following early ecological changes associated with hunting and scavenging, as well as a second period of adaptation after the rise of modern agriculture.

[1] Baker Institute for Animal Health, College of Veterinary Medicine, Cornell University, Ithaca, NY 14853, USA. [2] Department of Molecular Biology and Genetics, Cornell University, Ithaca, NY 14853, USA. [3] Department of Biomedical Sciences, College of Veterinary Medicine, Cornell University, Ithaca, NY 14853, USA. ✉email: jjl336@cornell.edu; dankoc@gmail.com

Infectious diseases have been a major source of selective pressure during the evolution of modern humans[1–3]. A handful of select case studies have revealed evidence of geographically localized positive selection for alleles that provide resistance to endemic infectious diseases, like malaria[4] and Lassa fever[5]. Yet despite abundant evidence that infectious diseases have repeatedly driven human adaptation[6–8], we understand relatively little about how exposure to new pathogens can shape host genetics.

*Bacillus anthracis* is a bacterium responsible for anthrax disease[9,10]. *B. anthracis* primarily afflicts ruminants during the course of its natural life-cycle and has driven selective pressures in cattle and sheep[11,12]. While anthrax disease is believed to rarely affect most primate species, with exceptions[13,14], anthrax disease has been a notable source of mortality in humans[9,10]. Genetic diversity of extant *Bacillus* strains suggests that anthrax-causing *Bacillus* pathogens evolved in sub-Saharan Africa[15,16], likely well before humans migrated around the globe[17]. *B. anthracis* then radiated from Africa into the middle east and Europe ~3000–6000 years ago in the mid-Holocene, possibly following the spread of Neolithic agricultural practices[16]. *B. anthracis* strains endemic to Europe later spread around the globe concurrent with increased trade and colonization[16]. Thus, although anthrax disease has been associated with the rise of animal husbandry[18–20], the putative African origin of anthrax disease-associated *Bacillus* species may have provided opportunities for early humans to encounter *B. anthracis*, or the ancestral *B. cereus*, while hunting or scavenging ruminant game species well before the advent of modern farming.

Here, we investigate the transcriptional landscape of anthrax toxin receptor 2 (*ANTXR2*), the membrane receptor granting anthrax toxins access to host cells. We find evidence that sequential ecological and behavioral changes drove recurrent adaptation in the complex regulatory landscape around *ANTXR2*. Our results are consistent with at least two stages of human adaptation to anthrax disease—first to early ecological changes associated with hunting and scavenging, followed by the second period of adaptation after the rise of agriculture.

## Results

### Changes in *ANTXR2* transcription and mRNA in early human ancestors

Anthrax infection produces toxins that target immune cells to dampen the immune response and aid in spore migration throughout the body[21]. A key component of the adaptive immune system, CD4 + T cells, has measurable changes in activation, cytokine production, chemotaxis, and differentiation after anthrax infection[22–24]. To identify evolutionary differences associated with anthrax susceptibility in humans, we studied changes in CD4 + T cells between humans and non-human primates. We confirmed that anthrax toxins inhibit T-cell activation using enzyme-linked immunosorbent assay (ELISA) for the T-cell activation marker IL-2 (Supplementary Fig. 1). We then studied transcriptional changes between human and non-human primate CD4 + T cells using PRO-seq (Pol II loading) and RNA-seq (mature mRNA). We focused our analysis on the *ANTXR2* gene, which encodes a ubiquitously expressed transmembrane receptor that aids in the cellular entry of toxins secreted by the *B. anthracis* bacterium[25–27] (Fig. 1a). Consistent with our hypothesis, *ANTXR2* was transcribed at eightfold lower levels in human CD4 + T cells compared to non-human primates at the level of both Pol II loading (chimpanzee and rhesus macaque) and mRNA (rhesus macaque and baboon) ($P < 0.0001$) (Fig. 1b, c and Supplementary Figs. 2a, b and 3). Expanding our analysis of RNA-seq data from CD4 + T cells to 91 humans confirmed the loss of *ANTXR2* expression in all of the available human data[28]

(Fig. 1d and Supplementary Fig. 4). Under the assumption of maximum parsimony, our comparative analysis indicates that *ANTXR2* was downregulated in the human lineage.

We examined whether expression changes in *ANTXR2* were specific to CD4 + T cells. We found that *ANTXR2* was expressed at significantly lower levels in CD14 + monocytes isolated from humans than from rhesus macaque or baboon ($P < 0.05$) (Supplementary Fig. 5). Expanding this analysis using published RNA-seq from multiple immune cell types in humans[28] and rhesus macaque[29] identified ~10-fold lower *ANTXR2* expression in the majority of human immune cell types, including B cells, CD8 + T cells, and monocytes (Supplementary Fig. 6). Differences in *ANTXR2* expression between species were not identified in natural killer cells, although this cell type had relatively low levels of *ANTXR2* mRNA in both species. Next, we asked whether changes in *ANTXR2* expression were specific to the immune system using microarray data[30] comparing three tissues (heart, kidney, and liver) in human, chimpanzee, and rhesus macaque. We found no consistent human-specific decrease in *ANTXR2* expression in these other tissues, suggesting that decreased expression of *ANTXR2* in humans is restricted to the immune system (Supplementary Fig. 7). Collectively, analysis of *ANTXR2* expression data was consistent with our hypothesis of ancient evolutionary changes affecting *ANTXR2* expression in the immune system during the early divergence of humans from other primates.

To more precisely date the occurrence of expression changes, we analyzed RNA-seq data from peripheral blood mononuclear cells (PBMC), a cell population largely composed of immune cells, in multiple human populations. We found that *ANTXR2* expression was lower in two African populations, including Batwa, a hunter gatherer population historically in the Great Lakes region of Africa, and Bakiga, agriculturalists from neighboring Rwanda and Uganda, compared to rhesus macaque[31,32] (Supplementary Fig. 8). We also noted substantial variability within human populations, including a higher *ANTXR2* expression in hunter-gatherer than agrarian populations (examined in detail, below). These results suggest that expression differences between human and rhesus macaque began before human populations expanded and migrated out of Africa.

To confirm that *ANTXR2* expression changes of the magnitude observed between human and non-human primates can affect sensitivity to anthrax toxins, we overexpressed *ANTXR2* using CRISPR activation (CRISPRa)[33]. We selected K562 cells as a model system because K562 is a human hematopoietic cell line that could be manipulated in culture and showed a basal level of *ANTXR2* expression. CRISPRa increased *ANTXR2* mRNA levels in K562 by tenfold relative to an empty vector control 24 h after transfection, a change that was similar in magnitude to the difference observed between humans and non-human primates (Fig. 1e). *ANTXR2* overexpressing cells had a significantly lower viability following treatment with recombinant anthrax toxins: protective antigen (PA) and FP59, a potent analog of lethal factor (LF)[34] (Fig. 1f). Thus a tenfold increase in human *ANTXR2* expression, similar to changes found in non-human primates, increased the effect of anthrax toxins on cellular phenotypes and confirmed that *ANTXR2* expression causes changes in anthrax toxin sensitivity in human hematopoietic cells.

### Independent DNA sequence differences distributed in multiple CREs underlie *ANTXR2* expression changes

We asked whether cis-regulatory changes could account for transcriptional divergence in *ANTXR2* between species. We set out to identify cis-regulatory elements (CREs) near the *ANTXR2* locus in primates.

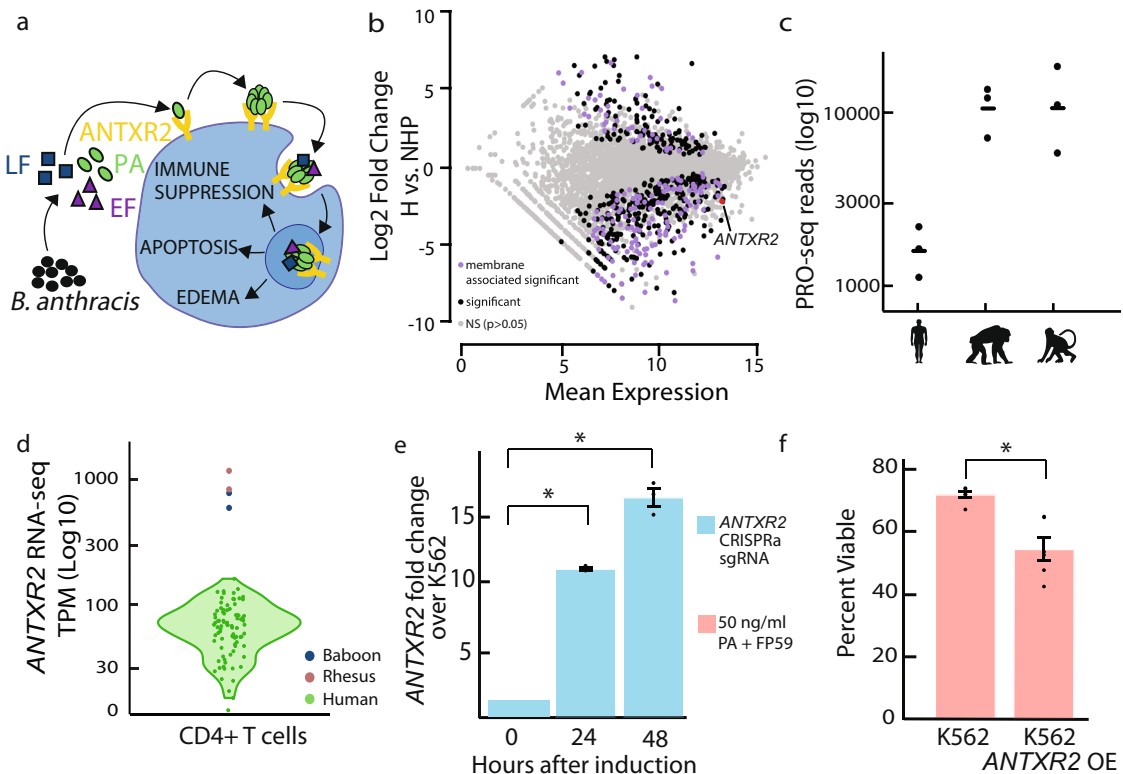

**Fig. 1 Decreased *ANTXR2* in humans can affect anthrax toxin sensitivity in blood cells. a** *B. anthracis* toxins lethal factor (LF) and edema factor (EF) cause apoptosis, edema, and immune suppression in target host cells. **b** Differentially transcribed protein-coding genes between humans (H) and non-human primates (NHP) in CD4 + T cells based on PRO-seq data (differential expression based on DESeq2, *P* < 0.0001, multiple testing correction made). The GO term "integral component of the membrane" is enriched in differentially transcribed genes, as seen in purple. *ANTXR2* is highlighted in red. **c**) CD4 + T-cell PRO-seq data shows that *ANTXR2* is transcribed eightfold lower in human than chimpanzee and rhesus macaque (*P* < 0.0001, DESeq2, multiple testing correction made). **d** Comparison of *ANTXR2* RNA-seq expression levels in CD4 + T cells among a large set of humans (*n* = 91) to rhesus macaque and baboon. Human variation in expression does not overlap with rhesus macaque or baboon expression. Human individuals are primarily self-reported as white or Asian (see "Methods" for details). **e** CRISPRa induction in K562 cells results in significantly increased *ANTXR2* expression after 24 and 48 h (*P* = 0.007, one-sided *t* test) measured using qRT-PCR. Data are presented as mean values + /− SEM. *n* = 3 independent CRISPRa inductions. Source data are provided as a Source Data file. **f** K562 cells that overexpress endogenous ANTXR2 are significantly less viable after an anthrax toxin challenge as measured by the Alamar blue viability stain (*P* = 0.002, one-sided *t* test, SE error bars). Data are presented as mean values + /− SEM. *n* = 6 independent CRISPRa inductions over two independent experiments. Source data are provided as a Source Data file.

We used 12 transcription initiation regions previously identified in CD4 + T cells from human, chimpanzee, and rhesus macaque using dREG[35,36], a tool that identifies regulatory elements using patterns of nascent transcription in PRO-seq data (see "Methods"). A comparison with DNase-I hypersensitivity and H3K27ac ChIP-seq data in human CD4 + T cells did not identify additional candidate CREs in this locus (Supplementary Fig. 9), indicating that dREG was sensitive enough to pick out the most important candidate CREs. The 12 candidate CREs were spread throughout the *ANTXR2* locus, including inside of the *ANTXR2* transcription unit, situated between *ANTXR2* and the neighboring protein-coding gene, *PRDM8*, and within the region surrounding the *PRDM8* transcription unit (Fig. 2a).

Different species rarely show major, coordinated changes in the activity of candidate CREs within a locus, and the exceptions are associated with major changes in gene expression and the appearance of new traits[37,38]. Testing each putative CRE for differential transcription, a hallmark of regulatory activity at enhancers[39], revealed a strong bias for decreased transcription in the human lineage across the *ANTXR2* locus (median 2.3-fold lower transcription in human; *P* = 0.001; Wilcoxon rank-sum test; Supplementary Fig. 10). At least three CREs were differentially transcribed on the human lineage using a conservative test for differential transcription (DESeq2[40] FDR corrected *P* value < 0.05),

all of which were down-regulated. The significant degree of *cis*-regulatory divergence between human and non-human primates at multiple annotated CREs is uncharacteristic of randomly selected genomic regions. Moreover, multiple changes at distinct CREs suggest that *cis*-acting loci, rather than *trans*-acting factors, underlie much of the divergence in *ANTXR2* expression.

While species-specific variation in *ANTXR2*-associated CREs is suggestive of CRE-mediated transcriptional evolution, we aimed to explicitly link distal CREs to the *ANTXR2* gene. To verify that annotated CREs interact with the *ANTXR2* promoter, we performed in situ Hi-C[41] and Micro-C[42,43] in CD4 + T cells from human and rhesus macaque and tested for the enrichment of contacts in the *ANTXR2* locus (Supplementary Table 1). Our Hi-C data revealed that *ANTXR2*, *PRDM8*, and all candidate CREs were found within the same topological associated domain (TAD), a structure reported to insulate distal enhancers from affecting expression[44,45] (Fig. 2b). Moreover, Hi-C data revealed focal contacts between the *ANTXR2* promoter and candidate CREs, especially those situated between the *ANTXR2* and *PRDM8* genes (Fig. 2b and Supplementary Fig. 11). The main focal contacts observed in human Hi-C data had a higher normalized contact frequency with the *ANTXR2* promoter in rhesus macaque, potentially reflecting species-specific differences in *ANTXR2* transcription (Fig. 2c and Supplementary Fig. 12).

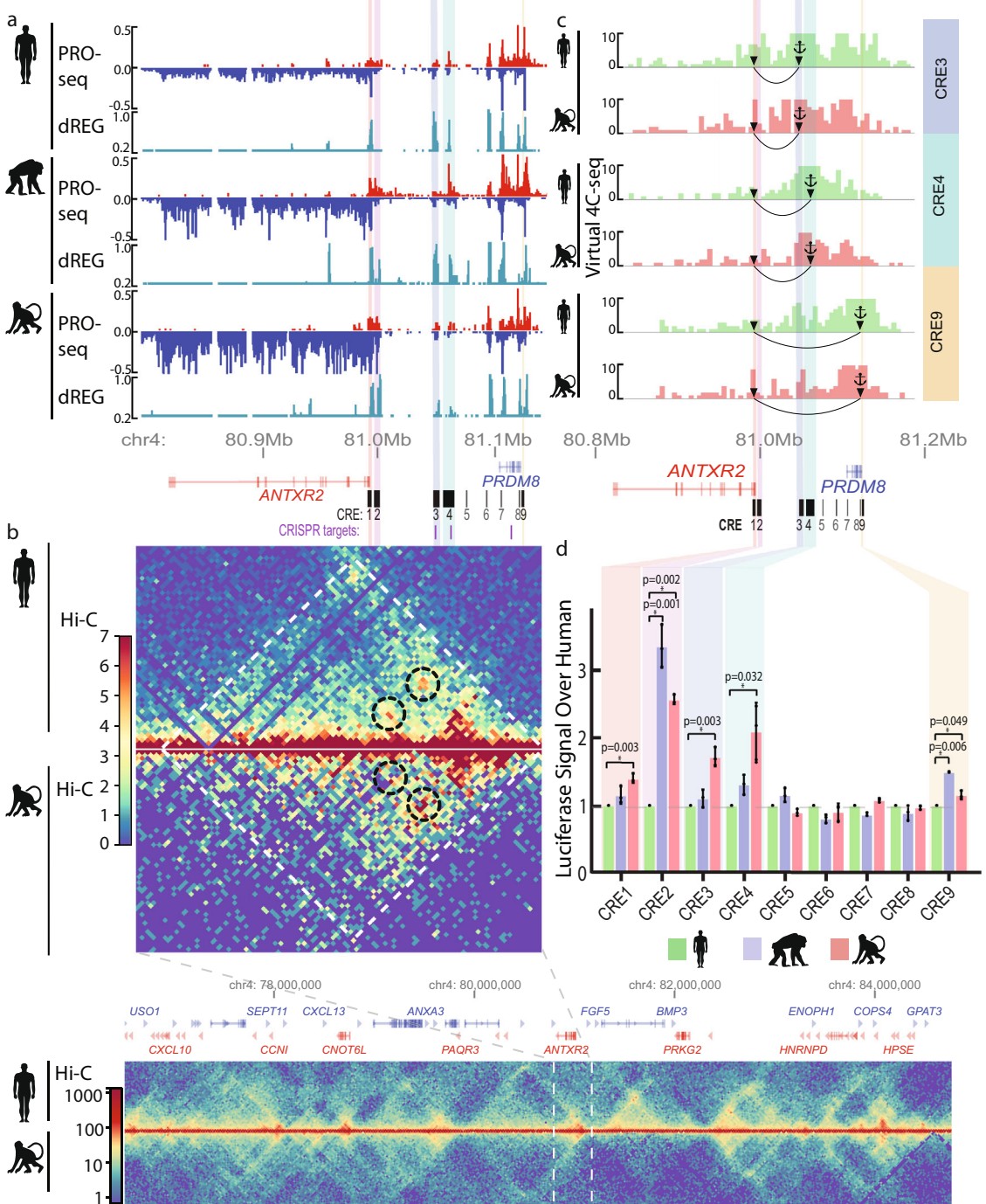

**Fig. 2 Changes in *ANTXR2 cis*-regulatory element activity and chromatin structure. a** Genome browser shot of CD4 + T-cell PRO-seq (normalized by reads per million, plus strand in red, minus strand in blue) in human, chimpanzee, and rhesus macaque. Regulatory elements predicted by dREG are shown in teal. *Cis*-regulatory elements (CREs) tested in this study (CRE1-9) are shown below the data tracks. CREs having decreased activity in human (as determined by luciferase in (**d**)) are shaded. Overlap with validated CRISPR targets[46] that result in reduced *ANTXR2* expression are shown below the CREs in purple. **b** Hi-C/Micro-C in CD4 + T cells from human and rhesus macaque. Normalized contacts are shown in the heatmap scale to the left. *ANTXR2* is located in the same topological domain (TAD) as the upstream gene *PRDM8* in both species. TADs are marked with white dotted lines. Focal contacts with the *ANTXR2* promoter are circled with black dotted lines. Zoomed-out gene-level Hi-C contacts are pictured below, with the region shown in the top panel outlined in white. **c** Virtual 4C-seq signal calculated based on the number of Hi-C contacts for CREs tested in the luciferase assay. CREs with decreased activity in human are colored. The anchor symbol at each CRE represents the bait region that all contacts are derived from. The arrow at the end of the arc denotes the *ANTXR2* promoter. **d** Luciferase assay performed in Jurkat cells to test the activity of regulatory elements in human, chimpanzee, and rhesus macaque shows lower activity in human compared with either chimpanzee or rhesus macaque for CRE1, CRE2, CRE3, CRE4, and CRE9. Data are presented as mean values + / − SEM. One-sided *t* tested used and significant *P* values are shown in the figure. *n* = 3 independent transfections in each species. Source data are provided as a Source Data file.

At least two other lines of evidence indicate that CREs identified here affect *ANTXR2* expression. First, we identified *ANTXR2* expression quantitative trait loci (eQTL) that overlapped CREs between *ANTXR2* and *PRDM8* using RNA-seq data from human CD4 + T cells[28] (Supplementary Fig. 13). Second, using CRISPRi to knock down two of the CREs identified here (CRE3 and 4) in a K562 model system[46] decreased *ANTXR2* expression by 45% and 62%, respectively (Fig. 2a). Taken together, these findings indicate that the CREs identified here, most notably those found in between *ANTXR2* and *PRDM8*, affect *ANTXR2* expression in CD4 + T cells.

Multiple CREs changed Pol II loading in human samples, which raised the possibility that independent genetic changes in multiple CREs contributed to changes in *ANTXR2* expression. To determine whether changes in the transcriptional activity at multiple CREs reflect independent DNA sequence changes, we used a luciferase assay to measure the enhancer activity of CREs in DNA sequences obtained from human, chimpanzee, and rhesus macaque. We focused our experiments on 9 of the 12 candidate CREs, which we selected for evolutionary conservation between chimpanzee and rhesus macaque. In addition, the nine CREs were situated in the *ANTXR2* promoter, or in the region upstream of the *ANTXR2* transcription start site, that are likely to affect *ANTXR2* expression in the native genomic context based on our contact analysis, eQTL, and CRISPRi evidence (noted above). We used Jurkat human leukemic CD4 + T cells as a model trans-environment which recapitulates the pattern of transcription in the *ANTXR2* locus observed in primary T cells better than alternative models, including K562 (Supplementary Fig. 14). Five of the nine candidate CREs showed higher luciferase activity in chimpanzee or rhesus macaque than in human ($P < 0.05$, t test; Fig. 2d). Decreased overall activity in five of nine CREs, with no CREs increased in humans, is unlikely to happen by chance ($P = 0.03$, binomial test for human loss/no-change; $P = 0.01$, multinomial test for human gain, loss, or no change, assuming 36% of CREs change[47]). Previous studies have noted multiple, compensatory changes in the activity of sequences within a CRE[47–50]. Consistent with this idea, using separate luciferase assays to analyze the *ANTXR2* promoter revealed compensatory changes in activity within distinct regions of the *ANTXR2* promoter (Supplementary Fig. 15). This shows that DNA sequences in the *ANTXR2* locus have continued to drift through time, consistent with other loci. However, this drift led to decreased function in the majority of CREs within the locus. Thus, we conclude that multiple genetic changes within the *ANTXR2* locus have decreased enhancer activity in the human lineage, collectively contributing to decreased *ANTXR2* expression. These independent genetic changes may reflect ancient adaptation for reduced *ANTXR2* expression in humans.

**Recent selective sweep in Europeans upstream of *ANTXR2*.** In addition to changes compared with other primates, *ANTXR2* expression varied by one order of magnitude within humans, suggesting that a number of genetic variants may still exist within human populations associated with anthrax susceptibility. Most extant *B. anthracis* strains outside of Africa show genetic similarity to those in Europe, leading to a model in which *B. anthracis* that was endemic to Europe during the Neolithic era spread throughout the globe alongside trade and colonization during the past several thousand years[16,51–53]. Previous reports have noted lower anthrax toxin sensitivity in Europeans[54], suggesting that European populations may be more resistant to endemic anthrax strains than non-European populations. To determine how selection has influenced modern human populations at the *ANTXR2* locus, we computed the composite-likelihood ratio

(CLR) of a selective sweep genome-wide in four human populations (Europeans [CEU], East Asians [CHB and JPT], and Africans [YRI]) using SweepFinder2[55]. Consistent with the hypothesis that anthrax disease in Europeans led to a relatively recent population-specific adaptive response, we found a candidate selective sweep upstream of *ANTXR2* (Fig. 3a). We speculate that the affected haplotype reflects a hard sweep, because SweepFinder2 and related tools are most sensitive to hard sweeps[56]. The selective sweep had a higher CLR in European (CEU) than 99.5% of other loci genome-wide (Supplementary Fig. 16). Moreover, the CLR was substantially higher in Europeans compared to 1000 Genomes populations representative of East Asian (CHB, JPT) or African (YRI) ancestry (Fig. 3a). The *ANTXR2* locus and SNPs within *ANTXR2* have shown up in previous scans for positive selection in the human genome[57,58], which provides additional evidence that the locus surrounding *ANTXR2* is under positive selection.

We computed nucleotide diversity (Pi) for the CEU, CHB, JPT, and YRI populations and found decreased diversity in the predicted selective sweep in CEU, CHB, and JPT compared to YRI (Fig. 3b). To account for differences in overall diversity levels between the populations, we analyzed where the average diversity value for the interval near *ANTXR2* fell in the genome-wide distribution. The average nucleotide diversity in the selective sweep interval fell within the following genome-wide percentiles: CEU = 1.87, CHB = 4.98, JPT = 4.36, and YRI = 51.09 (Supplementary Fig. 17). This sweep overlapped two CREs associated with decreased *ANTXR2* expression between humans and non-human primates (CRE2 and CRE3). Collectively, evidence of reduced *ANTXR2* expression in humans followed by further evolutionary modification of the *ANTXR2* locus in Europeans suggests a complex evolutionary history driven by recurrent selection around *ANTXR2*.

**Patterns of human variation in *ANTXR2* expression across tissues.** A recent sweep discovered by SweepFinder2 may affect *ANTXR2* expression in different ways than observed for the ancestral expression changes between human and non-human primates. Comparative analysis of RNA-seq and microarray data between human and non-human primates (presented above) indicated that changes in *ANTXR2* expression were primarily limited to the immune system (Fig. 1 and Supplementary Figs. 6 and 7). However, anthrax disease is a systemic disorder affecting multiple organs in different manners. Anthrax disease progresses with attacks on multiple organ systems, including cardiomyocytes and smooth muscle cells (major targets of the lethal toxin), as well as dermis, hepatocytes, and lungs (targets of edema toxin)[59,60]. We, therefore, asked whether more recent changes in *ANTXR2* expression between human populations were also restricted to the immune system.

To evaluate expression between different human populations, we divided GTEx RNA-seq data[61] into individuals who were confirmed to have some evidence of European or African ancestry based on the mitochondrial haplotype[62]. We found that individuals of African ancestry had higher *ANTXR2* expression, on average, compared with individuals of European ancestry, for the majority of tissues ($n = 22/27$) (Fig. 3d). Despite limited power to identify differences with relatively few samples that have evidence of African ancestry in GTEx, the change in muscle expression was statistically significant on its own (individuals of European ancestry/individuals of African ancestry log2 fold change of 0.52, FDR = 0.032 based on DESeq2). In addition, combining *P* values across tissues using Fisher's method also supported a significantly decreased *ANTXR2* expression in individuals of European ancestry in tissues overall ($P = 0.00015$,

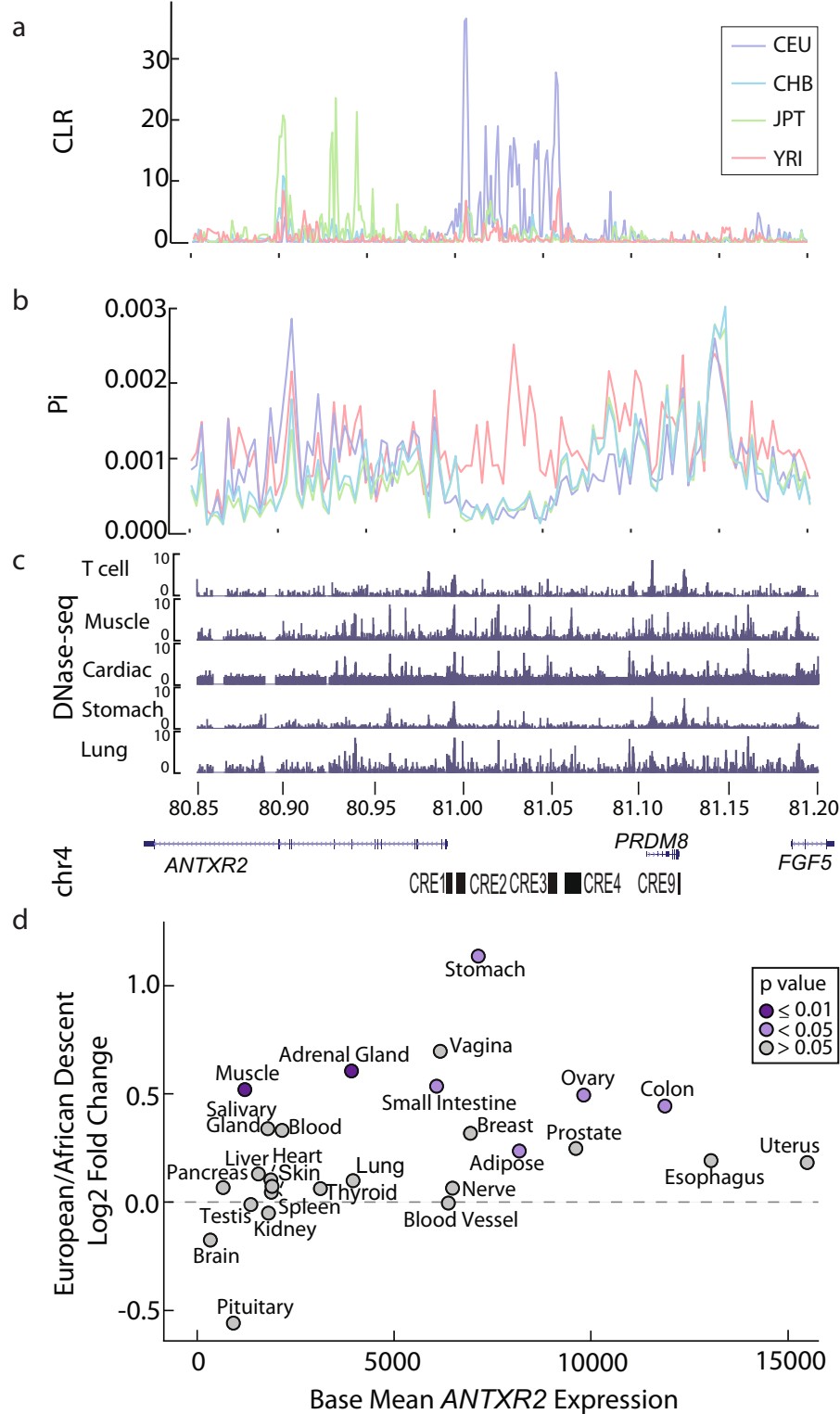

**Fig. 3 Selective sweep affects *ANTXR2* expression in multiple tissues. a** Sweepfinder2 CLR (composite-likelihood-ratio) values for CEU (Northern and Western Europe), CHB (China), JPT (Japan), and YRI (Yoruba) populations show a predicted selective sweep in the CEU population upstream of *ANTXR2* at 81–81.05 Mb. **b** Pi diversity values for CEU, CHB, JPT, and YRI populations show decreased diversity in the predicted selective sweep in CEU, CHB, and JPT (see Supplementary Fig. 17 for percentiles). **c** ENCODE DNase-I-seq data for T cells, muscle, heart, stomach, and lung show different numbers and locations of DNase-I hypersensitive sites, which indicates differential regulatory landscapes in cell types within the predicted selective sweep in CEU. **d** Differential expression of GTEx data for all available tissues showed a range of fold changes of individuals of European descent compared to individuals of African descent (as determined by mitochondrial haplotype). The majority of tissues showed greater expression in individuals of African descent ($n = 22/27$). Source data are provided as a Source Data file.

Fisher's method). Thus, our analysis identified population-level differences in *ANTXR2* expression affecting the majority of tissues, with a lower expression in Europeans as would be expected based on the likely direction of positive selection due to interactions with anthrax disease.

The activity of CREs is highly cell-type and tissue-specific. Therefore, CREs which were targets of more recent selective pressure may not be the same as those which were identified in CD4 + T cells. To identify a set of CRE candidates, we examined human DNase-I hypersensitivity data in 42 tissues from the Epigenome Roadmap project[63]. We found DNase-I hypersensitive sites from multiple tissues that overlapped with the interval of high CLR values (Fig. 3c and Supplementary Fig. 18). The number of DNase-I hypersensitive sites within the interval ranged from 0 to 17 in different tissue types. Muscle and cardiac tissue datasets were frequently outliers which had more DNase-I hypersensitive sites intersecting in the CLR region than other tissues (Supplementary Fig. 19). This finding is perhaps noteworthy because apoptosis caused by lethal toxins in muscle tissue, especially the heart, is often the cause of host death in anthrax disease[64,65]. Thus, we conclude that more recent selection in European populations may have influenced *ANTXR2* expression in multiple tissues susceptible to apoptosis or edema by targeting DNase-I hypersensitive sites within the genomic region affected by the candidate-selective sweep.

**Patterns of genetic differentiation supports continued human divergence upstream of *ANTXR2*.** Finding a putative selective sweep specific to Europeans led us to ask whether the *ANTXR2* locus differentiates CEU from other human populations. To address this, we calculated relative population differentiation (Fst) between Europeans (represented by CEU) and either East Asians (represented by CHB and JPT) or Africans (represented by YRI). The entire *ANTXR2* locus showed elevated differentiation between European and non-European populations, with a median value of 0.14–0.33 (64–96th percentile genome-wide) (Fig. 4a). The greatest signal of differentiation occurred outside of the putative selective sweep but overlapped upstream CREs (including CRE2) that diverged between human and non-human primates (Fig. 4b). One particularly interesting haplotype, directly adjacent to the region of high CLR, had a very high Fst in Europeans relative to all other ethnic groups. This region included several SNPs overlapping CREs upstream of *ANTXR2*, including rs41407844—the allele frequency of which was correlated with reported variation in anthrax toxin sensitivity in lymphoblastoid cell lines[54]: high frequency of the derived allele in Europeans (~0.85), intermediate in East Asians (~0.38), and low frequency in African populations (~0.17; Fig. 4c–e). This result, and the observation of elevated Fst at CRE3, CRE4, and CRE9 in the absence of high CLR profile, suggests that Europeans have maintained genetic separation at loci derived from both incomplete soft sweeps on genetic variation from early human divergence and a candidate hard sweep associated with recent European anthrax exposure (Supplementary Fig. 20).

## Discussion
Our work is consistent with the hypothesis that humans adapted to anthrax exposure during several periods of human evolution (Fig. 4f). First, ecological changes associated with increased hunting and scavenging early in the evolution of modern humans may have led to increased interactions with *Bacillus* species in Africa. Indeed, *Bacillus* species are believed to have originated in Africa[66], and remain endemic to Africa, where they are a source of mortality for wildlife, including wild chimpanzees[13,14]. It stands to reason that human ancestors who adopted hunting and

scavenging behaviors, especially of ruminant species in endemic areas, would have faced a higher burden of anthrax disease than neighboring chimpanzee relatives. Second, we found evidence of more recent selective pressures acting in Europeans that are consistent with historical records of recent outbreaks within Europe[18,52,67]. The nature of human interactions with anthrax disease might make the *ANTXR2* locus particularly susceptible to soft selective sweeps over time. Both prehistoric and modern anthrax exposure was likely highly sporadic and caste specific—dependent upon a variety of factors including regional endemic areas of the globe, the availability of ruminant species, and the roles of individuals within the population. Indeed, anthrax disease disproportionately affected agricultural and textile workers during the 19th century[10]. Moreover, *ANTXR2* expression was slightly higher in blood cells isolated from hunter gatherers than in nearby agricultural populations within Africa[31], potentially consistent with agricultural populations being at higher risk of exposure to anthrax disease. In summary, our findings imply a complex evolutionary history in the *ANTXR2* locus, supporting a model in which selection on multiple alleles has continued to drive changes in *ANTXR2* expression within human populations.

## Methods

**Human research participants**. We have complied with all relevant ethical regulations and informed consent was obtained from all participants. This work was approved by the Cornell University IRB under protocol 1506005662.

**Animal research**. This work was approved by the Cornell University IACUC under protocol 2009-0044. Welfare and handling of all primates were covered under IACUC protocols governing each individual primate center.

### Experimental methods
*Isolation of CD4 + T cells from humans and non-human primates.* Experiments using samples from humans and animals were done in compliance with Cornell University IRB and IACUC guidelines. We received peripheral blood samples (60–100 mL) from healthy adult humans, chimpanzees, rhesus macaques, and baboons. Human subjects gave informed consent for inclusion in this study. We used a minimum of three individuals from each primate species to account for differences in within-species variation in gene transcription. Blood was collected in purple top EDTA tubes and either shipped overnight on ice (non-human primate samples) or maintained at 4 °C (human samples) before processing. Blood was diluted 50:50 with PBS. To collect peripheral blood mononuclear cells (PBMCs), 20 mL of blood:PBS mixture was applied over 20 mL Ficoll-Pacque and centrifuged (750×g) for 30 min at 20 °C to create a gradient. Cells from the PBMC layer of the Ficoll gradient were collected and washed three times in ice-cold PBS. If red blood cell contamination occurred, the cell pellet was treated with ACK red blood cell lysis buffer. CD4 + T cells were isolated using CD4 microbeads (Miltenyi Biotech, 130-045-101 [human and chimp], 130-091-102 [rhesus macaque and baboon]). CD14 + T cells were isolated using CD4 microbeads (Miltenyi Biotech, 130-045-101 [human and chimp], 130-091-102 [rhesus macaque and baboon]). In total, 10^8 cells were resuspended in a binding buffer (PBS with 0.5% BSA and 2 mM EDTA). Cells were then bound to CD4 microbeads (20 μL of microbeads/10^7 cells) for 15 min at 4 °C. Cells were washed with 1–2 mL of PBS/BSA solution, resuspended in 500 μL of binding buffer, and passed over a MACS LS column (Miltenyi Biotech, 130-042-401) on a magnet at 4 °C. The MACS LS column was washed three times with 2 mL PBS/BSA solution, before being eluted off the magnet. Cells were counted using a hemocytometer, resuspended in RPMI-1600, and incubated at 37 °C for a minimum of an hour before further processing.

*Luciferase assays.* Genomic DNA was isolated using a Quick-DNA Miniprep Plus Kit (#D4068S; Zymo Research) following the manufacturer's instructions from human, chimp, and rhesus macaque PBMCs aliquots depleted for CD4 + cells. CRE regions were amplified from the genomic DNA, restriction digested with a combination of KpnI, SacI, or MluI (depending on primer region), and cloned into a pGL3-promoter vector (Promega). The same orthologous regions were amplified from all three species with identical primers where possible or species-specific primers covering orthologous DNA in diverged regions. The media (RPMI-1640) was changed in Jurkat cells one day before transfection. RPMI-1640 with 20% FBS was equilibrated in plates in a 37 °C incubator prior to transfection. On the day of transfection, Jurkat cells were centrifuged at 100×g for 10 min, washed with PBS, and centrifuged again. After centrifugation, Jurkat cells (2 million per reaction) were resuspended in 100 μL room temperature Mirus electroporation solution. Vectors were co-transfected with pRL-SV40 Renilla (Promega) in a 20:1 ratio (2 μg pGL3 to 100 ng pRL-SV40). Electroporation was done in a Lonza Nucleofector 2b

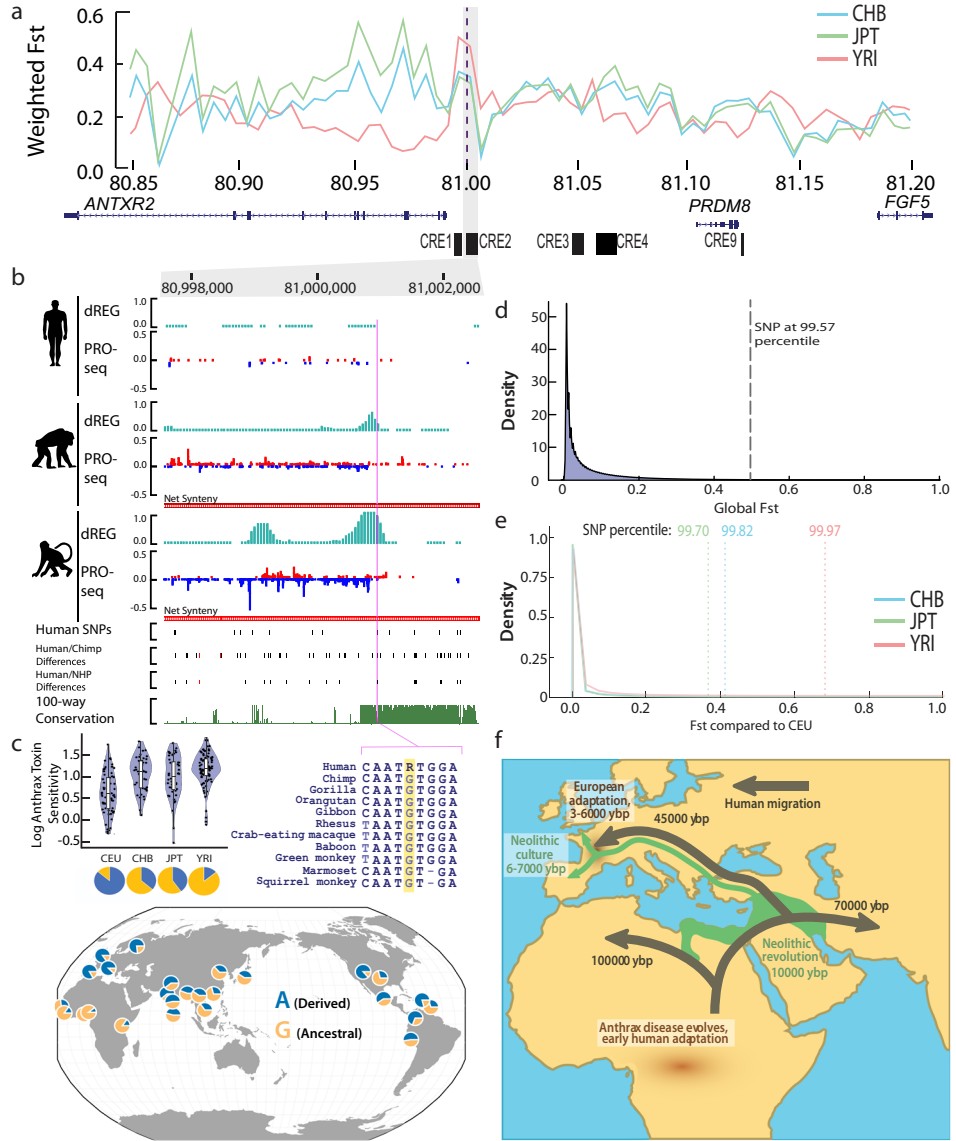

**Fig. 4 Genetic differentiation in European populations consistent with multiple phases of selection driving changes in *ANTXR2*. a** Weighted Fst (fixation index) comparisons between CEU (Northern and Western Europe) and CHB (China), JPT (Japan), and YRI (Yoruba) HapMap populations at the *ANTXR2* locus show a peak around *ANTXR2*. **b** PRO-seq (normalized by reads per million) and dREG signal (pictured in teal) at *cis*-regulatory element 2 (CRE2). rs41407844 falls within a dREG peak shared by chimpanzee and rhesus macaque and the ancestral allele G is conserved within the primate lineage. Conservation of CRE2 between humans and non-human primate tracks show human-specific single nucleotide polymorphisms (SNPs), human/ chimpanzee differences (SNPs in black, insertion/deletions (INDELS) in red), human/non-human primate differences (SNPs in black, INDELS in red), and 100-way conservation. Net synteny tracks show the position of regions that have one-to-one orthologs in the chimpanzee and rhesus macaque genomes. **c** Martchenko et al.[54] showed that CEU lymphoblastoid cell lines (LCLs) have lower sensitivity to anthrax toxins than LCLs derived from other populations. Data are presented as mean values + /− SD (CEU $n = 60$, CHB $n = 43$, JPT $n = 44$, YRI $n = 84$). The allele frequency of SNP rs41407844 across populations. Source data are provided as a Source Data file. **d** Genome-wide global Fst distribution[84]. The SNP rs41407844 is above the 99.57 percentile of Fst of all SNPs genome-wide. **e** Rs41407844 is above the 98% percentile for pairwise Fst comparisons between CEU and CHB, JPT, and YRI HapMap populations genome-wide. **f** Model demonstrating the pattern of human migration (dark green) compared to the spread of Neolithic culture (light green) and anthrax disease distribution (brown).

device using program X-001. Immediately after electroporation, 1 mL equilibrated media was added to the cuvette and then the cell mixture was added to six-well plates and incubated at 37 °C. Eighteen hours post-transfection, luminescence was measured in triplicate using the Dual-Luciferase® Reporter Assay System (Promega).

Luciferase assay primers—See Supplementary Table 2.

*CRISPRa in K562.* Use of K562 cell line: K562 was used for CRISPR experiments because of its ability to be transfected and manipulated in culture. K562 possesses basal expression of *ANTXR2*, allowing for comparison to CRISPR activation. A key component of the CRISPRa experiment is treating CRISPR manipulated cells with anthrax toxins to measure the effect of the expression on viability. K562 cells

remained viable and healthy enough after transfection to be used in toxin viability assays. Cell lines that more closely replicate the trans environment of human CD4 + T cells, such as Jurkat, were not viable for days after transfection.

*Generation of dCas9-KRAB K562 line:* Phoenix Hek cells (grown in DMEM) were seeded in a six-well plate at 400,000 cells/plate. Cells were grown until ~90% confluent and then 1 µg of pHAGE_EF1a_dCas9-KRAB plasmid from Addgene (#50919) plasmid was transfected using Lipofectamine 3000 (Invitrogen). Twenty-four hours later, 3 mL of the virus was mixed with 10 µg/mL polybrene and incubated for 5 min at room temperature. Virus plus polybrene was added to 300,000 K562 cells. The cells were centrifuged for 40 min at 800×*g* at 32 C. Twenty-four hours later, the virus was removed after centrifugation and fresh RPMI-1600 was added. Starting 24 h later, the cells were selected with 150 µg/mL Hygromycin

B for 2 weeks. The K562 dCas9-KRAB stable cell lines were grown and maintained in Hygromycin B and RPMI-1600.

*sgRNA cloning*: Primers for sgRNAs were designed using ChopChop (http://chopchop.cbu.uib.no/) for the *ANTXR2* gene and off-target controls. Primers were located -400 to −50 bp away from the TSS for CRISPRa. See primers ANTXR2_CRISPRa_for and ANTXR2_CRISPRa_rev in Supplementary Table 2 for sequence. A G was added at the 5' end of primers for use with a U6 promoter, along with restriction sites for cloning. Forward and reverse sgRNAs were synthesized separately by IDT and annealed. T4 Polynucleotide Kinase (NEB) was used to phosphorylate the forward and reverse sgRNA during the annealing. In total, 10 × T4 DNA Ligase Buffer, which contains 1 mM ATP, was incubated for 30 min at 37 °C and then at 95 °C for 5 min, decreasing by 5 °C every 1 min until 25 °C. Oligos were diluted 1:200 using Molecular grade water. sgRNAs were inserted into the pLenti SpBsmBI sgRNA Hygro plasmid from Addgene (#62205) by following the author's protocol (26501517). The plasmid was linearized using BsmBI digestion (NEB) and purified using gel extraction (QIAquick Gel Extraction Kit). The purified linear plasmid was then dephosphorylated using Alkaline Phosphatase Calf Intestinal (CIP) (NEB) to ensure the linear plasmid did not ligate with itself. A second gel extraction was used as before to purify the linearized plasmid. The purified dephosphorylated linear plasmid and phosphorylated annealed oligos were ligated together using the Quick Ligation Kit (NEB). The ligated product was transformed into One Shot Stbl3 Chemically Competent *E. coli* (ThermoFisher Scientific). In total, 100 μL of the transformed bacteria were plated on Ampicillin (200 μg/mL) plates. Single colonies were picked, sequenced, and the plasmid was isolated using endo-free midi-preps from Omega.

*Transfection of sgRNA plasmid*: The day prior to transfection, the media (RPMI-1640 with 10% FBS) was changed in K562 cells, and cells were diluted to a concentration of 1 million cells/mL. On the day of transfection, K562 cells were centrifuged at 100×g for 10 min, washed with PBS, and centrifuged again. RPMI-1640 with 10% FBS was equilibrated in plates in a 37 °C incubator prior to transfection. After centrifugation, K562 cells (1 million per reaction) were resuspended in 100 μL room temperature Mirus electroporation solution. In total, 2 μg of the sgRNA plasmid was added to each reaction. Electroporation was done in a Lonza Nucleofector 2b device using the program for K562 cells. Immediately after electroporation, 1 mL equilibrated media was added to the cuvette and then the cell mixture was added to six-well plates and incubated at 37 °C. Twelve hours after transfection, 2 μg/mL of doxycycline was added to cells to activate dCas9-KRAB expression. To confirm the overexpression of *ANTXR2*, 4 h after the addition of doxycycline a portion of the cells were collected for RNA extraction using Trizol. cDNA was generated from RNA samples using the Thermo Fisher High Capacity RNA-to-cDNA kit and qPCR was performed using SsoAdvanced Universal SYBR Green master mix with primers to assay *ANTXR2* expression. See primers ANTXR2_qRT-PCR_for and ANTXR2_qRT-PCR_rev in Supplementary Table 2 for sequence.

**Toxin viability assays**. K562 cells transfected with CRISPRa plasmids were confirmed to have overexpression of *ANTXR2*. After confirmation and within the 12 h half-life window of the activating doxycycline, 50,000 K562 cells were added to wells of 96-well plates in 100 μL RPMI-1640, supplemented with 10% RPMI. Anthrax toxin PA (List Biological Laboratories 171D) was added to wells at a concentration of 1 μg/mL (or vehicle control). FP59, a recombinant anthrax lethal factor fused to the Pseudomonas Exotoxin A Catalytic Domain (Kerafast ENH013), which is capable of killing blood cells, was added at a concentration of 50 ng/mL (or vehicle control). Twenty hours after the addition of PA and FP59, 10 μL of Alamar Blue (Thermo Fisher #DAL1025) was added to each well. Four hours after the addition of Alamar Blue, fluorescence was measured using a plate reader with an excitation of 570 and an emission of 610.

**Activation of CD4 + T cells**. CD4 + T cells were isolated using the above procedure for two human subjects. After equilibration in RPMI-1640 with 10% FBS, cells were stimulated with 25 ng/mL PMA and 1 mM Ionomycin (P/I or π) or vehicle control (2.5 μL EtOH and 1.66 μL DMSO in 10 mL of culture media). Thirty minutes after activation or addition of the vehicle control, cells were treated with 2.5 μg/mL Recombinant PA (List Biological Laboratories 171D) and 500 ng/mL Recombinant LF (List Biological Laboratories 172A) or vehicle control. Twenty-four hours later, media from non-activated, non-activated with toxin treatment, activated, and activated with toxin treatment wells were collected for ELISA.

**IL-2 ELISA on CD4 + T cells**. ELISA was done using the R&D Human IL-2 DuoSet Kit (Catalog #DY202).

Plate preparation—16 h prior to the ELISA experiment, the capture antibody was added to 96-well plates at a concentration of 0.5 μg/μL in 100 μL of PBS. Plates were sealed and left at room temperature. The following day, the diluted capture antibody was aspirated and washed with 400 μL Wash Buffer three times. In total, 300 μL of Block Buffer was added to each well and incubated for at least an hour. After incubation, plates were washed with 400 μL Wash Buffer.

ELISA assay—100 μL of RPMI from the samples or standards diluted in the Reagent Diluent were added to the prepared 96-well plate. The plate was covered and left to incubate for 2 h at room temperature. In total, 100 μL of the Detection Antibody diluted at 1:60 with the Reagent Diluent was added to each well. The plate was washed with Wash Buffer. In all, 100 μL of streptavidin HRP diluted at 1:40 in the Reagent Diluent was added to each well. The plate was covered, protected from light, and incubated for 20 min at room temperature. The plate was washed and 100 μL of the Substrate Solution was added to each well, the plate was covered, protected from light, and incubated for 20 min at room temperature. In total, 50 μL of Stop Solution was added to each well and mixed. Fluorescence was measured on a plate reader at 450 nm and wavelength corrected at 540 nm. The relative signal was calculated using a standard curve for all samples. Samples activated with PMA and Ionomycin (see above) were compared to unactivated samples to get a fold change metric of activation.

**Hi-C and Micro-C library preparation**. Cell preparation—CD4 + T cells were isolated according to the above procedure. After >1 h of equilibration in RPMI-1640 supplemented with 10% FBS, cells were centrifuged at 300×g, washed with PBS, and centrifuged again. Cells were resuspended in a mixture of 1% formaldehyde in 1× PBS. Cells were incubated at room temperature for 10 min on a rocker. Formaldehyde was quenched by the addition of 2.5 M glycine to a Cf = 0.2 M. Cells were incubated for room temperature for 5 min on a rocker. Cells were centrifuged at 4 °C, washed in cold PBS, centrifuged, and PBS was aspirated. Pellets were flash-frozen using dry ice and stored at −80 °C prior to library preparation.

Hi-C—The protocol detailed in Rao et al.[41] was followed with the following adjustments. After the addition of the lysis buffer, cells were incubated on ice for 30 min. MboI (NEB #R0147) was used for restriction digestion. Following DNA purification, unligated biotin was removed using a mixture of 0.5 μL of 10 mM dATP, 0.5 μL dGTP, 20 μL 3000 U/mL T4 DNA polymerase (NEB #M02030) for each sample. Samples were incubated for 4 h at room temperature and then T4 DNA polymerase was inactivated at 72 °C for 20 min. Shearing was done using a Bioruptor sonicator using the LOW setting 30 S ON/ 90 S OFF for two cycles of 10 min. Libraries were prepared with the NEBNext Ultra II Library Preparation Kit (NEB #E7103). Samples were sequenced on a combination of Illumina's NovaSeq 6000 and HiSeq 4000 at Novogene.

Micro-C—For MNase digestion, cells were thawed on ice for 5 min, incubated with 1 mL MB#1 buffer (10 mM Tris-HCl, pH 7.5, 50 mM NaCl, 5 mM MgCl$_2$, 1 mM CaCl$_2$, 0.2% NP-40, 1× Roche cOmplete EDTA-free (Roche diagnostics, 04693132001) and washed twice with MB#1 buffer. MNase concentration for each cell type was predetermined as the concentration that gives ~90% mononucleosomes in an MNase titration experiment, using human CD14 + monocytes a CD4 + T cells, ranging between 2.5 and 20 U/million cells in a final reaction volume of 1 mL/million cells. Chromatin was digested with MNase for 10 min in 37 °C, and digestion was stopped by adding 8 μL of 500 mM EGTA and incubating at 65 °C for 10 min.

Following dephosphorylation with rSAP (NEB #M0371) and end polishing using T4 PNK (NEB #M0201), DNA polymerase Klenow fragment (NEB #M0210), and biotinylated dATP and dCTP (Jena Bioscience #NU-835-BIO14-S and #NU-809-BIOX-S, respectively), ligation was performed in a final volume of 2.5 mL for 3 h in RT using T4 DNA ligase (NEB #M0202). The dangling end was removed by a 5 min incubation with Exonuclease III (NEB #0206) at 37 °C and biotin enrichment was done using 20 μL Dynabeads™ MyOne™ Streptavidin C1 beads (Invitrogen #65001). Libraries were prepared with the NEBNext Ultra II Library Preparation Kit (NEB #E7103). Samples were sequenced on a combination of Illumina's NovaSeq 6000 and HiSeq 4000 at Novogene.

## Data analysis

*Hi-C and Micro-C analysis and visualization*. Individual Hi-C samples, combined with Micro-C samples, were mapped to hg19 (for human samples) or rheMac10 (for rhesus macaque samples) using Juicer[68] (version 1.6). Juicer was run using the standard command

 juicer.sh -z genome.fa.gz -p chrom.sizes -y/species_MboI.txt -s noneMboI

Replicates for each species were combined using Juicer's mega function. The combined rhesus macaque aligned dataset was lifted over to hg19 using Crossmap[69]. The inter_30.hic files were used as input for visualization. The .hic files were converted to mcool files and visualized using the HiGlass tool[70]. For virtual 4C-seq plots, reads from the merged_nodups.txt files generated from Juicer were used to get all contacts that connected to the promoter of *ANTXR2* and visualized on the genome browser.

*Mapping orthologs between species*. Cross-species comparison of genomic coordinates and genes was based on the methods in Danko et al.[38]. This strategy aligns all reads to the best available reference genome and converts the coordinates between each species genome to a human (hg19) reference system for comparison using CrossMap[69] (version 0.4.2). We used chain files that will conservatively only map 1:1 orthologies. Chain files were either sourced from the UCSC genome browser (https://hgdownload.cse.ucsc.edu/goldenpath/hg19/) or were produced in-house using scripts provided as part of the Kent source toolkit[71] (version 1.04.00). When counting reads to perform quantitative comparisons between different species, we also implemented several other filters in which we excluded positions that did not have a 1:1 orthology in the reciprocal best chain files or which are not uniquely mappable at 50 bp read length, in any of the species in the comparison. These filters ensure that we are counting exactly the same orthologous positions in all species that we compare.

*PRO-seq and RNA-seq differential expression.* PRO-seq data from human, chimpanzee, and rhesus macaque were published in Danko et al.[38] (GEO accession code GSE85337, https://www.ncbi.nlm.nih.gov/geo/query/acc.cgi?acc=GSE85337) and analysis of these data was done using methods adapted from that paper. The PRO-seq mapping pipeline removes reads that fail Illumina quality filters and trims adapters using cutadapt with a 10% error rate. Reads are then mapped with BWA[72] (version 0.7.17) to hg19, panTro4, or rheMac10. Mapped reads are converted to bigWig format for analysis using BedTools[73] (version 2.29.2) and the bed-GraphToBigWig program in the Kent Source software package version 1.04.00[74]. Chimpanzee and rhesus macaque PRO-seq data are then converted to hg19 using Crossmap[69] (see above) before using DESeq2[40] (version 1.26.0) to find differentially expressed genes.

*Predicting CREs using dREG.* We previously used dREG to identify candidate CREs using PRO-seq data from human, chimpanzee, and rhesus macaque CD4 + T cells[38]. dREG uses a machine learning strategy called support vector regression to score genomic sites based on patterns of PRO-seq data learned from the transcription start sites in a different cell type[35,36]. In Danko et al.[38], scores were computed on the combined signal for all individuals to maximize statistical power. dREG scores in chimpanzee and rhesus macaque were converted to hg19 coordinates using CrossMap and their corresponding reciprocal-best nets. The false discovery rate was set to be <7%, which corresponds to a dREG score threshold of 0.3. Here, we used dREG sites near *ANTXR2* from Danko et al. Because sites of divergently initiating Pol II identified by dREG, called transcription initiation regions, frequently act in concert with other nearby initiation regions[75,76], we merged candidate dREG regions within 1500 bp using BEDTools. After merging all candidate CREs, we had 14 candidates within the *ANTXR2* locus for analysis.

*ANTXR2 RNA-seq analysis in PBMCs.* Bakiga and Batwa PBMC RNA-seq data[31,32] were downloaded for the control condition (GEO series GSE120502). Rhesus macaque RNA-seq data were downloaded for the control condition (NCBI Bio-Project PRJNA246101). RNA-seq data were mapped using Salmon[77] (version 0.14.1) and NCBI RefSeq genes to hg19. Rhesus macaque RNA-seq data were mapped to hg19 using the same parameters. *ANTXR2* transcripts per million were compared between samples for transcript NM_001145794.1.

*Salmon reciprocal comparison.* Reciprocal salmon analyses were performed by mapping to both human/rhesus macaque genomes and both human/baboon genomes for a subset of samples to test the validity of performing Salmon (version 0.14.1) on non-human primate data using the human genome. Both genomes were mapped to each genome and the ratios were compared between species for expression (see Supplementary Fig. 4). For mapping to hg19, transcript NM_001145794.1 was used for all species. Transcript XM_015138456.2 was used to map to rheMac10 and transcript XM_021938471.1 was used to map to panAnu4. There are similar ratios between the reciprocal analyses (H/R = 0.16 mapped to hg19, H/R = 0.14 mapped to rheMac10) and (H/B = 0.21 mapped to hg19, H/B = 0.10 mapped to papanu4) that suggest mapping to hg19 may be a conservative estimate of the true difference in expression. Thus, we were comfortable with using hg19 for all Salmon analyses.

*DICE RNA-seq and eQTL analysis.* RNA-seq from 91 human CD4 + naive T-cell samples from DICE[28] was downloaded under dbGap protocol 23187, project phs001703.v1.p1. For other cell types (CD8 + T naive T cells, B cells, monocytes, and NK cells), 50 randomly selected samples were downloaded from DICE[28]. RNA-seq data were mapped using Salmon[77] (version 0.14.1) and NCBI RefSeq genes to hg19. Rhesus macaque RNA-seq data were mapped to hg19 using the same parameters. *ANTXR2* transcripts per million were compared between samples for transcript NM_001145794.1. We retrieved eQTLs for *ANTXR2*, as well as their effect size on *ANTXR2* expression, from the DICE online database and calculated pairwise LD scores between all *ANTXR2*-associated eQTLs using PLINK.

*RNA-seq analysis across GTEx tissues.* GTEx version 8 raw read counts per gene data (GTEx_Analysis_2017-06-05_v8_RNASeQCv1.1.9_gene_reads.gct.gz) was downloaded from the GTEx portal (https://gtexportal.org/home/datasets), along with the relevant, publicly available samples and subjects metadata files. mtDNA haplogroups were assigned to each sample[62] and samples of non-European or African origin were filtered out. The remaining samples were split based on tissue of origin. For each tissue, differential expression was calculated genome-wide between samples of European vs. African origin using DESeq2. P values for *ANTXR2* are reported.

*Multi-species RNA-seq analysis in multiple immune cell types.* All available Rhesus macaque RNA-seq from CD8 + T cells, CD4 + T cells, B cells, monocytes, and NK cells were downloaded from the Snyder-Mackler[29] dataset, GEO series GSE83302. RNA-seq data were mapped using Salmon[77] (version 0.14.1) and NCBI RefSeq genes to hg19. *ANTXR2* transcripts per million were compared between samples for transcript NM_001145794.1. Comparisons in expression to human samples of the same cell type (from DICE[28]) were done using a one-sided Wilcox rank-sum test.

*Microarray comparison between primate tissues.* Expression microarray data in human, chimpanzee, and rhesus macaque heart, liver, and kidney from Blekhman et al.[30] was downloaded via GEO series GSE11560. We analyzed signal intensity in the seven microarray probes for *ANTXR2* across six individuals in each species and used Blekman et al.'s pre-computed pairwise differential expression metric for comparing *ANTXR2* expression.

*Sweepfinder2 CLR scan.* Human genome data were taken from phase3 of the 1000 Genomes Project[78]. VCFs were subsetted based on their population group. The b-value maps that are used to compute the effect of background selection on the human genome were taken from[79]. Recombination maps from the deCODE database[80] were used. Sweepfinder2[55] (version 1.0) was used to calculate the composite-likelihood ratio for CEU, JPT, CHB, and YRI genome-wide. To calculate percentile for the region of the candidate selective sweep, average CLR values for non-overlapping 50 kb bins were calculated genome-wide and compared to the average value for the 50 kb selective sweep region.

*Positive selection summary statistics.* We obtained genome-wide estimates of nucleotide diversity in 30 kb bins from Pybus et al.[81]. To be able to assess how extreme a statistic was for given genomic loci we obtained a score, P as 100 minus the genome-wide percentile of the statistic divided by 100. Thus, generating a value bounded between 0 and 1 analogous to P value, where lower scores mean this statistic is more extreme in the genome-wide distribution. To calculate percentile for the region of the candidate selective sweep, average Pi values for non-overlapping 50 kb bins were calculated genome-wide and compared to the average value for the 50 kb selective sweep region.

We obtained global Fst estimates for each SNP genome-wide from Pybus et al.[81]. We calculated the distribution of global Fst of all SNPs and then found the percentile of SNP rs41407844.

All Fst values were computed using VCFtools[82] (version 0.1.16) using the --weir-fst command with a window size of 5 kb and a step size of 5 kb. In addition to the Fst calculations for bins of 5 kb, Fst was calculated for each SNP genome-wide. Fst was calculated for each SNP in all pairwise comparisons of CEU, YRI, JPT, and CHB and used to find the percentile for rs41407844 compared to genome-wide SNP values.

*Comparison of DNase-I-seq profiles across ENCODE tissues.* All DNase-I-seq datasets available from human tissues were downloaded from ENCODE (https://www.encodeproject.org/). Individual access used and links to download them can be found in the Source Data file. BedTools[73] was used to extract the number of DNase-I peaks within the 5 kb window of elevated CLR upstream of *ANTXR2*. Accessions shown in the genome browser view of Fig. 3c are ENCFF195KNE (T cell), ENCFF331SYD (lung), ENCFF704WQC (muscle), ENCFF990HMA (cardiac), and ENCFF490TJL (stomach). Representative peaks are shown in Supplementary Fig. 18.

*Other data used.* CD4 + T-cell H3K27ac was downloaded from the GEO series GSE40668. CD4 + primary T-cell DNase-I-seq data was from Epigenome Roadmap[63], downloaded from GEO GSM736592. Jurkat PRO-seq data were downloaded from GEO GSE66031. K562 PRO-seq data were downloaded from GEO GSE60456.

**Reporting summary**. Further information on research design is available in the Nature Research Reporting Summary linked to this article.

## Data availability

The rhesus macaque Hi-C, Micro-C, and RNA-seq and the baboon RNA-seq data generated in this study have been deposited in the GEO database under accession code GSE156161. The processed data are also available under accession code GSE156161. The human Hi-C, Micro-C, and RNA-seq data generated in this study have been deposited in dbGaP under project number phs002146.v1.p1, access can be obtained by requesting access to dbGaP. The CRISPRa, luciferase, and other functional assay data generated in this study are provided in the Source Data file and at https://github.com/Danko-Lab/ANTXR2. The PRO-seq data used in this study are available in the GEO database under accession code GSE85337. The DICE human immune cell RNA-seq data used in this study are available in the dbGaP database under project number phs001703.v1.p1, access can be obtained by requesting access to dbGaP. The rhesus macaque immune cell RNA-seq data used in this study are available in the GEO database under accession code GSE83302. The cross-species expression microarray data used in this study are available in the GEO database under accession code GSE11560. The Bakiga and Batwa population RNA-seq data used in this study are available in the GEO database under accession code GSE120502. Rhesus macaque PBMC RNA-seq data used in this study are available in the BioProject database under accession code PRJNA246101. Human DNase-1-seq data used in this study are available in the ENCODE database [https://www.encodeproject.org/], individual accession codes can be found in the Source Data file. The human tissue RNA-seq data used in this study are available from GTEx (https://gtexportal.org/home/datasets), The CD4 + T-cell H3K27ac data used in this study are available in the GEO database under accession code GSE40668. The CD4 + primary T-cell DNase-I-seq data

used in this study are available in the GEO database under accession code GSM736592. The Jurkat PRO-seq data used in this study are available in the GEO database under accession code GSE66031. The K562 PRO-seq data used in this study are available in the GEO database under accession code GSE60456. Source data are provided with this paper.

## Code availability

Scripts used for data analysis can be accessed at https://doi.org/10.5281/zenodo.5172835[83].

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

## Acknowledgements

We thank A. Siepel, H. Hijazi, A. Clark, B. Marks, and all members of the Danko lab for valuable discussions and suggestions. Work in this publication was supported by R01-HG010346 and R01-HG009309 (NHGRI) to CGD and an F31-1AI140050 (NIAID) to LAC. The content is solely the responsibility of the authors and does not necessarily represent the official views of the US National Institutes of Health.

## Author contributions

L.A.C., G.B., P.W.M. and E.J.R. performed experiments. L.A.C., G.B., I.S., P.R.M., J.J.L. and C.G.D. analyzed the data. C.G.D. and J.J.L. supervised data collection and analysis. L.A.C. and C.G.D. wrote the paper with input from the other authors.

## Competing interests

The authors declare no competing interests.
