## [Peer Review File · Nature Communications]

Title: Multiple stages of evolutionary change in anthrax toxin receptor expression in humansREVIEWER COMMENTS

Reviewer #1 (Remarks to the Author):

In the manuscript “Multiple stages of evolutionary change in anthrax toxin receptor in humans” Choate et al explore the evolutionary adaptation of a human gene responsible for sensitivity to anthrax disease. They evaluate gene expression changes, regulatory activity, 3D genome structure and selective sweeps around the genomic locus and find evidence of two selections that led to modern humans’ decreased susceptibility to anthrax. The work leverages publicly available datasets and provides new RNA-seq and HiC experiments, as well as performs several in vitro assays.

This is a compelling example of adaptation at the level of regulatory changes and gene expression that would be of interest to many readers. However, the presentation of the methods, data, figures and results could be significantly improved and more clearly described.

Major comments:

1. The data generated by this study “have been deposited in GEO, and an accession number will be added shortly” (page 17). The data generated should be publicly available, or privately but with access for reviewers, at the time of submission.

2. Three cis-regulatory regions between ANTXR2 and PRDM8 in the human genome have been functionally validated with CRISPR and are known to have an effect on ANTXR2 expression. They are included in the NCBI RefSeq database (see also Gasperini et. al. A Genome-wide Framework for Mapping Gene Regulation via Cellular Genetic Screens. Cell 2019). These should be analysed to see if they overlap with the CREs identified in this study. If so, this could help strengthen the results.

3. It would make the experiment and the results more understandable if there was one schematic or figure which unified all the main findings reported in the paper across this locus. For example, the identified CREs, their conservation, relevant 3D genome contacts, selective sweep regions and mutations. These are currently spread across different figures and it is difficult to connect how they relate to each other. There is also no figure which marks the genomic intervals considered as “gene deserts”, “surrounding” PRDM9, etc. which are often referred to in the text and could potentially be interpreted in different ways. There is also no figure clearly showing which CREs are differently transcribed in human (Results pg. 6, Figure S5) which could be marked on this figure.

4. Although the most parsimonious explanation of reduced expression in human compared to two other primates (Figure 1C, S3) is a loss in the human lineages, the underlying assumption and reasoning used should be clearly stated. Similarly, the use of the word “loss” (page 4) cannot be justified when referring to analyses involving only two species (Figure 1D, S2) because without a third species as an outgroup it is impossible to determine whether this was an evolutionary loss or gain. This should be clarified in the text and, if possible, third species analysis included.

5. On page 4 “dREG annotated 14 candidate CREs” – the parameters, p-value cut-offs, etc. for running dREG are not described anywhere in the paper. dREG is a software which predicts cis-regulatory elements from PRO-seq, but to the best of my knowledge does not categorise them. The classification of one of the CREs as a “broad promoter” should be described and explained? The numbers are also a little confused – there are 14 candidate CREs quoted in the results text, but only 9 are marked on Figure 2A and similar figures. Perhaps the 14 is a summation across all species?

6. Figure S4 shows DNase-I hypersensitivity and H3K27ac ChIP-seq in human. However, there is no explanation regarding where this data is sourced from or which tissues the experiments were performed on? It is possible that different tissues could explain why some identified CREs do not overlap H3K27ac or DNase peaks and this should be clarified because if they are active they should be in open chromatin in the same tissue.

7. There seems to be some methods confusion about Crossmap. Crossmap is an assembly conversion tool rather than a whole-genome aligner which is standardly used to convert between assemblies of the same species. It can be used to map between species using additional chain files sourced elsewhere. If this was done, the details about which chain files were used and their source should be included. Given the importance of whole genome alignment for the results of this study, most notably activity in orthologous positions between species, some validation of the alignments/cross-species mapping accuracy should be provided for example by showing the locations of the ANTXR2 and PRDM8 genes across species, including any gaps. Additionally, some analyses were performed on the older rheMac3 genome version and others on the newer rheMac8 which has considerably improved contiguity and thus both of these assemblies should be included in this validation.

8. For the Hi-C experiments, there are no details about how deeply the libraries were sequenced, how many reads remained after quality filtering, which options and result files were used with Juicer, and what was the resulting resolution? This information is crucial for interpreting the results and is missing.

9. For Figure 3C and related analyses and conclusions, all the tissues shown have reduced expression in European populations which leads the authors to conclude about selection in the tissues most affected by anthrax. However, to make the claim that this selection is tissue-specific, a negative control must be included. This could be a tissue anthrax does not act on and an effective control would be no difference in expression. As presented and without better justification the claims of tissue-specific adaptation are unsupported and should be removed and replaced with species- or population-specific adaptations.

10. ANTXR2 and DICE RNA-seq analyses were performed by mapping macaque and human RNA-seq to a human mRNA using Salmon and default mapping parameters, and consequently those mappings used to compare expression levels between the species. Given the divergence between species at the level of DNA, validation that mapping RNA-seq to another species transcript is not biasing the results should be provided.

Minor comments:

- It would be helpful to have line numbering for revisions.
- Pg. 3: malaria should not be capitalised
- Pg. 3: It would be helpful to explain which period is “3-6ka”.
- Which version of NCBI RefSeq was used for the annotation? The most current version includes a processed pseudogene in human between ANTXR2 and PRDM8 which might affect both the transcription seen and definitions of “gene deserts”
- Figure 1A, S11B – what do the track colors represent? In PRO-seq tracks blue seems to denote the negative strand, but the same color is also used in the dREG track in seemingly positive direction.
- Figure 1C and S7 – what is shown on the y axis?
- Figure 2B, S6 – what are the units on the color scale? It would also be helpful to have the genes and CREs annotated here.
- Pg. 6 - “majority of CREs in the gene desert”. Specific numbers should be provided.
- Figure 1C,S7 – what are the arrows and anchor? This information should be provided in the legend or figure.
- Figure S8 – what is the legend referring to (namely: TFH,TH17,TH1)?
- Pg. 6 “the sweep had minimal overlap” – how is “minimal” defined? What is the expectation? The sweep overlaps 2/9 CREs.
- Figure 4B and S11 – what does “Net Synteny” represent?
- Pg. 16 – the end of “Mapping orthologs between species” – is this section missing a word of a method/software at the end?

Reviewer #2 (Remarks to the Author):

Choate et al., hypothesized that human may have adapted to anthrax disease during several periods of human evolution. To test that hypothesis they combined population genetic tools with functional data to look for evidence for selection in the ANTXR2 gene locus. Their results suggest that there has been multiple waves of positive selection in ANTXR2, which are associated with lower expression of the gene. The paper is well written, the results are exciting and novel, and most conclusions are well supported by the data presented. My main critique is that often cases it feels that the data used was based on availability rather than something specifically designed to address the questions being posed. This does not necessarily diminish my overall enthusiasm for the study but it raises a number of questions that I think should be addressed before publication.

Specific comments

1. The first section of the Results is focused on the comparison of ANTXR2 expression levels in CD4 T cells between humans and Rhesus macaques. From a biological perspective, is there a reason to focus on CD4 T cells? Some rationale should be provided. In addition, the author talks about differences in ANTXR2 expression between humans and non-human primates but only data for Rhesus macaques are

shown. The fact that they see higher expression levels in Rhesus macaques as compared to humans cannot be interpreted as representative of most non-human primates. The author should at the very least include the expression levels of ANTXR2 in Chimpanzee for which expression data in CD4 T cells is also available.

2. In Figure 1D the authors compare expression levels of ANTXR2 between a panel of 85 humans and two rhesus. It would be extremely informative to expand the analyses to include additional rhesus samples and immune cell types. For example, the authors could use the data from the paper by Snyder-Mackler et al. Science (PMID: 27885030) for which data on 5 immune cell types from a panel of 45 Rhesus macaques is available. Knowing in what immune cell types we see a difference in ANTXR2 gene expression levels would help inform on the putative mechanisms that have been selected for.

3. The authors provide several lines of evidence that the genetic diversity patterns around the ANTXR2 locus are not compatible with neutral evolution, particularly among European populations. To my surprise, however, the authors did not try to connect the signals of selection observed to the eQTL for ANTXR2 present in the region. The expectation is that the allele(s)/haplotype(s) associated with lower gene expression would be the ones showing increased allele frequency in European populations. Is that the case? Any evidence for selection for these eQTL based on haplotype-based tests like iHS?

4. The authors talk about the fact that selection might have acted on regulatory elements that are specific to tissues like heart, lung and liver. There has been several comparative studies of gene expression in these tissues between human, rhesus and chimps (e.g., Blekhman et al, PLoS Genetics, PMID: 19023414, among several others). It would be important to show that in these tissues ANTXR2 shows lower expression levels in humans as compared to other primates (the expectation based on their proposed evolutionary model).

Reviewer #3 (Remarks to the Author):

This paper presents a novel, thorough and very compelling investigation of the function and evolution of anthrax susceptibility in humans, but I have concerns about the later part of the paper, which focuses on within-human evolution. The results are not sufficient to support the conclusion of a significant difference in Europeans, let alone to conclude that natural selection, rather than random drift or other population events, is responsible. This is, of course, a particularly fraught topic given that it involves comparing Europeans to Africans and Asians, and needs to be handled with great care.

Overall, I really enjoyed reading this manuscript, and for the first half have minor only comments. I was incredibly impressed by the functional investigation of the ANTXR2 locus, which really took the genomic story of divergence between species and figured out what was happening in terms of genome function. The authors' choice to start with the functional analysis at this locus was very effective, and drew me

into the science.

In the results section, it gets a bit confusing to follow which comparisons are happening where, and adding a bit more specificity would help, including more details in the figures. For example, in the first paragraph of the results, the authors refer to comparing human CD4+ T-cells to 85 humans, but some information on the ancestry of those 85 humans seems relevant given the hypothesis of evolution in the paper. I'm also unclear which comparisons were significant - the authors refer to the 8-fold downregulation in humans compared to non-human primates, but there are no measures of statistical significance in either Figure 1C or Fig S2B. In contrast, figure S3 has a statistical analysis, but doesn't highlight what appears to be a much more substantial difference between both human populations and rhesus macaque than between the human populations, or give a measure of magnitude - it just states that all the differences are significant. In the second paragraph of results, the authors switch from CD4+ t-cells, which they had justified using as a model, to K562 cells, but don't provide any description/justification for using that cell line in the main text. In figure 1E and 1F (despite using much more informative box plots in the supplemental figures) the authors present bar charts, and don't show either the individual data points or even give details on the number of samples. It's not shown if any of the comparisons in 1E are significant.

In the multispecies comparison (Figure 2), it's not clear whether the authors looked for regulatory elements present in the primates and missing in the humans, or just at elements conserved in humans. . In figure 2A, I don't know what the red and green vertical shading means. Is it somehow human and macaque like is shown in the key in panel 2C? In Figure 2C, there is no explanation of what the y axis is. I'm not sure how that panel demonstrates "the main focal contacts observed in human Hi-C data had a higher contact frequency ... ", or if there is a difference between the three CRE. In figure 2D, the authors chose to use a bar plot rather than a boxplot (maybe this is standard - I'm not sure) and didn't show sample points or give sample numbers. In 2D, the non-significant comparisons are not shown, which bothers me - I'd rather see all the results. In 2D, I'm not sure why Jurkat cells are now being used, and not T-cells or K562 cells. I'm very sure there is a good reason, but the manuscript would be more readable with a bit more context scientists less familiar with this type of assay.

The statements about expected conservation and divergence of CREs feels very vague. Is there any way to measure how significant the "significant degree of cis-regulatory divergence" between human and non-human primates. is? E.g. how often would such a pattern of conservation in macaque/chimpanzee & divergence in humans be expected in the genome, given what we know of CRE conservation.

Minor points In 3rd paragraph, "dReg" is used without any explanation of what that is. The word "notably" is used, twice. Particularly in the first usage, there isn't any context to understand why the following observation is notable.

My concerns start with the section on the recent selective sweep in Europeans. It seemed very odd to me that the authors presented their own, new, analysis limited to just one chromosome, using the 1000G project data. Selection scans should only be done in the context of the whole genome, in order to

show how unusual a locus is, given all the patterns that other population events can leave in the genome. Furthermore, many other groups have already done genomewide scans using exactly the same datasets used in this paper. The authors should show the results from those previous studies for the ANTXR2 locus. If those studies did not conclude there was evidence for positive selection at this locus, why not?

The authors then look at ANTXR2 gene expression. This paragraph is frustratingly vague. The information from Roadmap feels entirely descriptive. For example - "We found substantial numbers of DNase-I hypersensitive sites overlapping the interval of high CLR values in heart, lung, liver, and dermis". I have no idea how interesting this really is. Across the genome, there are lots of DNase-I hypersensitive sites in heart, lung, liver, and dermis, many of which will not be found in blood, and lots of intervals in the genome with high CLR values. The authors need to somehow demonstrate that there is something unusual about this locus.

The evidence of increased expression in African individuals (figure 4C) is weak and problematic. First, the authors refer both to "African" (in the key) and "African American" (legend), which are obviously not interchangeable populations. Second, "African" is not usually used at all to refer to populations, given the massive diversity within that continent. If these individuals were in fact African American, and African refers simply to have some degree of African ancestry (which would be the case if based purely on mitochondrial genotypes) the range of African ancestry, and what this means for the analysis, should be specified. Finally, and most problematically, there is no significant difference in expression even on the single locus level, and no information on whether the pattern seen at this locus is unusual in the context of the whole genome.

The FST analysis is also unconvincing. FST is an incredibly noisy statistic, and I could not tell from the analysis whether any of the patterns observed at ANTXR2 are anything more than random noise.

We have now carefully revised our manuscript in response to the thoughtful and constructive comments and criticisms that were raised by our three anonymous reviewers. We were very happy to see that reviewers agreed that this example of evolutionary adaptation at the molecular level was thoroughly investigated and is of broad interest (“compelling”, “novel”). We found the thoughtful and constructive comments made by reviewers to be extremely helpful in preparing our revisions. In response, we have made substantial revisions, including the addition of new original data and the analysis of additional public datasets. We believe that our revised manuscript has improved substantially, and will be read with interest by a wide audience in the evolution, gene regulation, genomics, and immunology communities.

We note that all genomic data are now publicly available in either GEO or dbGap. We have also included a supplementary table, an excel file, that includes all raw data data from locus-specific experiments. Scripts used for analysis can now be found on Github. Finally, we have updated all figures to include individual data points.

Below please find a full point-by-point response addressing the reviewer’s comments in detail.

REVIEWER COMMENTS AND SUMMARY

Reviewer #1:

In the manuscript “Multiple stages of evolutionary change in anthrax toxin receptor in humans” Choate et al explore the evolutionary adaptation of a human gene responsible for sensitivity to anthrax disease. They evaluate gene expression changes, regulatory activity, 3D genome structure and selective sweeps around the genomic locus and find evidence of two selections that led to modern humans’ decreased susceptibility to anthrax. The work leverages publicly available datasets and provides new RNA-seq and HiC experiments, as well as performs several in vitro assays.

This is a compelling example of adaptation at the level of regulatory changes and gene expression that would be of interest to many readers. However, the presentation of the methods, data, figures and results could be significantly improved and more clearly described.

Response: We are grateful to the reviewer for their interest in our manuscript. We also appreciate the time the review took to make such constructive and useful comments. We believe that we have fully addressed the comments raised by the reviewer, as outlined in the point-by-point response, below.

Major comments:

1. The data generated by this study “have been deposited in GEO, and an accession number will be added shortly” (page 17). The data generated should be publicly available, or privately but with access for reviewers, at the time of submission.

Response: We fully agree with the importance of sharing data rapidly after posting a preprint. New genomic data can be found under GEO under accession number GSE156161 (Rhesus Hi-C and Micro-C raw data and processed data) and in dbGaP under project number phs002146.v1.p1 (human Hi-C and Micro-C raw data). All accession number IDs have been added to the manuscript and are available without restriction.

2. Three cis-regulatory regions between *ANTXR2* and *PRDM8* in the human genome have been functionally validated with CRISPR and are known to have an effect on *ANTXR2* expression. They are included in the NCBI RefSeq database (see also Gasparini et. al. A Genome-wide Framework for Mapping Gene Regulation via Cellular Genetic Screens. Cell 2019). These should be analysed to see if they overlap with the CREs identified in this study. If so, this could help strengthen the results.

Response: Two of the three regions validated by Gasparini et. al. (2019) with CRISPRi in K562 cells do indeed overlap CREs identified in our study. We agree that these results strengthen the evidence that these CREs affect *ANTXR2* expression, and we are grateful to the reviewer for pointing out the existence of these datasets. We have included these regions in Figure 2A and discuss them in the main text (pp. 5):

Independent DNA sequence differences distributed in multiple CREs underlie *ANTXR2* expression changes

Second, using CRISPRi to knock-down two of the CREs identified here (CRE3 and 4) in a K562 model system⁴⁶ decreased *ANTXR2* expression by 45% and 62%, respectively (**Fig. 2A**).

3. It would make the experiment and the results more understandable if there was one schematic or figure which unified all the main findings reported in the paper across this locus. For example, the identified CREs, their conservation, relevant 3D genome contacts, selective sweep regions and mutations. These are currently spread across different figures and it is difficult to connect how they relate to each other. There is also no figure which marks the genomic intervals considered as “gene deserts”, “surrounding” *PRDM9*, etc. which are often referred to in the text and could potentially be interpreted in different ways. There is also no figure clearly showing which CREs are differently transcribed in human (Results pg. 6, Figure S5) which could be marked on this figure.

Response: We have included a new supplementary figure summarizing the key findings from our paper (**Fig. S20**), including global Fst, PRO-seq, dREG signal, 3D contacts, conservation in mammals, CREs, luciferase results, selective sweep signatures, region descriptions, and fold

change of all dREG elements within the regions. We agree that this figure is helpful to visualize the different sources of data that were integrated in the present study. In addition, we think this figure is also useful because it juxtaposes the different regions affected by candidate markers for population-specific or cross-species selective pressure.

Supplementary Figure 20. Summary of findings at the *ANTXR2* locus. **A)** Global F_{st} values for all SNPs around the *ANTXR2* locus. **B)** Genome browser shot of CD4+ T-cell PRO-seq (normalized by reads per million, plus strand in red, minus strand in blue) in human and rhesus macaque. Regulatory elements predicted by dREG are shown in teal. **C)** 100-way conservation in mammals. **D)** RefSeq genes, lncRNA, and GENCODE Version 37 Pseudogenes. **E)** CREs tested in this study (CRE1-9) are shown below the data tracks. CREs with lower activity in human are shaded. **F)** CREs with lower activity in human than in chimpanzee or rhesus macaque as determined by luciferase. **G)** Log₂ fold change of PRO-seq expression in humans compared to non-human primates for regulatory elements defined by Danko et al³⁸. **H)** Region of increased CLR in the CEU population.

4. Although the most parsimonious explanation of reduced expression in human compared to two other primates (Figure 1C, S3) is a loss in the human lineages, the underlying assumption and reasoning used should be clearly stated.

Response: We agree that it is important to state our assumptions about maximum parsimony clearly. In the revised manuscript, we clarified that we are using a maximum parsimony assumption when concluding that transcription of *ANTXR2* decreased in humans. Relevant changes to the main text include: (pp. 3, see text below next response):

Similarly, the use of the word “loss” (page 4) cannot be justified when referring to analyses involving only two species (Figure 1D, S2) because without a third species as an outgroup it is impossible to determine whether this was an evolutionary loss or gain. This should be clarified in the text and, if possible, third species analysis included.

Response: We fully agree with the reviewer that RNA-seq data, which does not have an outgroup, provides no direct evidence that lower *ANTXR2* mRNA reflects a human-specific loss. In the revised manuscript we avoid making the claim that changes in *ANTXR2* RNA-seq abundance between primates reflects a human loss.

Nevertheless, we do think that when considering both the PRO-seq and mRNA-seq datasets, a human loss is the most likely explanation for changes in *ANTXR2* expression between primates. The PRO-seq data from human, chimpanzee, and rhesus macaque under a maximum parsimony assumption, as described above, does suggest a human-specific loss. It therefore seems likely that similar changes between human, rhesus macaque, and newly added data from baboon also likely to reflect a loss of expression in human. We have revised the text to clarify the basis for our conclusions in this section (pp. 3), stating that:

Changes in *ANTXR2* transcription and mRNA in early human ancestors

Consistent with our hypothesis, *ANTXR2* was transcribed at 8-fold lower levels in human CD4+ T-cells compared to non-human primates at the level of both Pol II loading (chimpanzee and rhesus macaque) and mRNA (rhesus macaque and baboon) ($p < 0.0001$) (Fig. 1B-C; Fig. S2A-B; Fig. S3). Expanding our analysis of RNA-seq data from CD4+ T-cells to 91 humans confirmed the loss of *ANTXR2* expression in all of the

available human data²⁸ (Fig. 1D, Fig. S4). Under the assumption of maximum parsimony, our comparative analysis indicates that *ANTXR2* was down-regulated in the human lineage.

5. On page 4 “dREG annotated 14 candidate CREs” – the parameters, p-value cut-offs, etc. for running dREG are not described anywhere in the paper. dREG is a software which predicts cis-regulatory elements from PRO-seq, but to the best of my knowledge does not categorise them. The classification of one of the CREs as a “broad promoter” should be described and explained? The numbers are also a little confused – there are 14 candidate CREs quoted in the results text, but only 9 are marked on Figure 2A and similar figures. Perhaps the 14 is a summation across all species?

Response: We clarified how the 14 dREG sites correspond to the 9 CREs tested in the luciferase experiments in the revised manuscript. Briefly, we identified 14 independent dREG sites that were active in at least one species using data from Danko et al. (2018) Nature Eco Evo as a starting point for our analysis. We analyzed all 14 in the genomic experiments. From these 14 candidates, we selected 9 for testing in luciferase experiments based on evidence for conservation in chimpanzee/ rhesus, while also selecting sites that appeared to be interacting with the *ANTXR2* promoter based on our Hi-C and eQTL analysis. The revised manuscript numbers all 14 independent dREG sites in Fig. 2A. It then identified the subset of 9 that were tested in luciferase experiments, and described the rationale for those choices. Specifically, see the revised Methods section (pp.17):

Methods:

Predicting CREs using dREG

We previously used dREG to identify candidate CREs using PRO-seq data from human, chimpanzee, and rhesus macaque CD4+ T-cells³⁸. dREG uses a machine learning strategy called support vector regression to score genomic sites based on patterns of PRO-seq data learned from transcription start sites in a different T-cell type^{35,36}. In Danko et al. 2018³⁸, scores were computed on the combined signal for all individuals to maximize statistical power. dREG scores in chimpanzee and rhesus macaque were converted to hg19 coordinates using CrossMap and their corresponding reciprocal-best nets. The false discovery rate was set to be <7%, which corresponds to a dREG score threshold of 0.3. Here we used dREG sites near *ANTXR2* from our Danko et al. 2018. Because sites of divergently initiating Pol II identified by dREG, called transcription initiation regions, frequently act in concert with other nearby initiation regions^{75,76}, we merged candidate dREG regions within 1500 bp using BEDTOOLS. After merging all candidate CREs, we had 14 candidates within the *ANTXR2* locus for analysis.

Additionally, we removed or refined the descriptions of the categories of CREs indicated by the reviewer. We agree that many of these descriptions were subjective, and we have tried to be more precise in the revised text. See especially (main text [pp. 4]):

Independent DNA sequence differences distributed in multiple CREs underlie *ANTXR2* expression changes

We asked whether cis-regulatory changes could account for transcriptional divergence in *ANTXR2* between species. We set out to identify cis-regulatory elements (CREs) near the *ANTXR2* locus in primates. We used 12 transcription initiation regions previously identified in CD4+ T-cells from human, chimpanzee, and rhesus macaque using dREG^{35,36}, a tool which identifies regulatory elements using patterns of nascent transcription in PRO-seq data (see Methods). A comparison with DNase-I hypersensitivity and H3K27ac ChIP-seq data in

human CD4+ T-cells did not identify additional candidate CREs in this locus (Fig. S9), indicating that dREG was sensitive enough to pick out the most important candidate CREs. The 12 candidate CREs were spread throughout the *ANTXR2* locus, including inside of the *ANTXR2* transcription unit, situated between *ANTXR2* and the neighboring protein-coding gene, *PRDM8*, and within the region surrounding the *PRDM8* transcription unit (Fig. 2A).

6. Figure S4 shows DNase-I hypersensitivity and H3K27ac ChIP-seq in human. However, there is no explanation regarding where this data is sourced from or which tissues the experiments were performed on?

Response: All data in Figure S4 were taken from human CD4+ T cells, collected by the Roadmap Epigenomics consortium. We have updated the methods section (pp. 19) and the main text (pp. 4, see below) to reflect the data sources.

Methods

Other data used

CD4+ T-cell H3K27ac was downloaded from GEO series GSE40668. CD4+ primary T-cell DNase-I-seq data was downloaded from Epigenome Roadmap⁶². Jurkat PRO-seq data was downloaded from GEO GSE66031. K562 PRO-seq data was downloaded from GEO GSE60456.

It is possible that different tissues could explain why some identified CREs do not overlap H3K27ac or DNase peaks and this should be clarified because if they are active they should be in open chromatin in the same tissue.

Response: To clarify, the primary goal of Fig. S4 was to convince readers that there weren't any additional candidate CREs we missed by considering only transcription, rather than chromatin accessibility or H3K27ac ChIP-seq data. As we expected, sites identified using dREG in human CD4+ T-cells were also accessible using DNase-I and had evidence for enriched H3K27ac in ENCODE data. We have tried to clarify the rationale for including this comparison in the revised manuscript, see especially (pp. 4):

Independent DNA sequence differences distributed in multiple CREs underlie *ANTXR2* expression changes

A comparison with DNase-I hypersensitivity and H3K27ac ChIP-seq data in human CD4+ T-cells did not identify additional candidate CREs in this locus (Fig. S9), indicating that dREG was sensitive enough to pick out the most important candidate CREs.

7. There seems to be some methods confusion about Crossmap. Crossmap is an assembly conversion tool rather than a whole-genome aligner which is standardly used to convert between assemblies of the same species. It can be used to map between species using additional chain files sourced elsewhere. If this was done, the details about which chain files were used and their source should be included. Given the importance of whole genome alignment for the results of this study, most notably activity in orthologous positions between species, some validation of the

alignments/cross-species mapping accuracy should be provided for example by showing the locations of the ANTXR2 and PRDM8 genes across species, including any gaps.

Response: We have clarified how we used CrossMap and where we obtained chain files from in the revised Methods section.

We used the same cross-species coordinate conversion strategy that we validated and described previously, in Danko et. al. *Nature Eco Evo* (2018). Briefly, this strategy aligns all reads to the best available reference genome and converts the coordinates between each species genome to a human (hg19) reference system for comparison using CrossMap. We used chain files that will conservatively only map 1:1 orthologies (these are known as reciprocal best chain files). Chain files were either sourced from the UCSC genome browser (<https://hgdownload.cse.ucsc.edu/goldenpath/hg19/>) or were produced in-house using scripts provided as part of the Kent source toolkit. When counting reads to perform quantitative comparisons between different species, we also implemented several other filters in which we excluded positions which did not have a 1:1 orthology in the reciprocal best chain files or which are not uniquely mappable at 50bp read length, in any of the species in the comparison. These filters ensure that we are counting exactly the same orthologous positions in all species that we compare. This strategy is advantageous, because it ensures that synteny (i.e., the order and orientation of protein coding genes and other highly conserved blocks) is preserved. This pipeline, and the rationale for it, is now described in additional detail in the Methods section (see Mapping orthologs between species, pp. 16):

Methods

Mapping orthologs between species

Cross-species comparison of genomic coordinates and genes was based on the methods in Danko et al. 2018³⁸. Briefly, this strategy aligns all reads to the best available reference genome and converts the coordinates between each species genome to a human (hg19) reference system for comparison using CrossMap⁶⁹. We used chain files that will conservatively only map 1:1 orthologies. Chain files were either sourced from the UCSC genome browser (<https://hgdownload.cse.ucsc.edu/goldenpath/hg19/>) or were produced in-house using scripts provided as part of the Kent source toolkit⁷¹. When counting reads to perform quantitative comparisons between different species, we also implemented several other filters in which we excluded positions which did not have a 1:1 orthology in the reciprocal best chain files or which are not uniquely mappable at 50bp read length, in any of the species in the comparison. These filters ensure that we are counting exactly the same orthologous positions in all species that we compare.

In addition, as requested by the reviewer, we show data at the orthologous locus in the uncovered coordinates system of each individual species (**Supplementary Fig. 3**).

Finally, we also note that we did not use crossmap in comparisons that leverage RNA-seq data, and our results are all consistent in these additional data modalities. We agree that consistency across analysis using independent analysis strategies is reassuring that our results are not confounded by technical issues.

Additionally, some analyses were performed on the older rheMac3 genome version and others on the newer rheMac8 which has considerably improved contiguity and thus both of these assemblies should be included in this validation.

Response: We have converted all analyses to the rheMac10 genome, using the reciprocal best chain files obtained from the UCSC genome browser, as noted in the revised methods section (see above).

8. For the Hi-C experiments, there are no details about how deeply the libraries were sequenced, how many reads remained after quality filtering, which options and result files were used with Juicer, and what was the resulting resolution? This information is crucial for interpreting the results and is missing.

Response: We have added **Supplementary Table 1**, which shows the sequencing statistics on the new *in situ* Hi-C data in the revised manuscript. All species were represented by multiple individuals and were sequenced to a depth of at least 500 million uniquely mappable contacts. For the purposes of this analysis, we combined all individuals within each species.

	Human hg19	Rhesus rheMac10
Sequenced read pairs	1537590113	962448103
Normal paired	878369068	733240864
Chimeric paired	436270983	135018624
Chimeric ambiguous	164587895	59713541
Unmapped	58362167	34475074
Alignable (normal+chimeric)	1314640051	868259488
Hi-C contacts	741551222	351837113
Inter-chromosomal	159541332	77932242
Intra-chromosomal	582009890	273904871
Short range (<20Kb)	342099230	175012244
Long range (>20Kb)	239059227	98875560

Supplementary Table 1. Hi-C and Micro-C mapping statistics. For each species, individuals were combined using the Juicer⁶⁸ pipeline. Human Hi-C contacts totalled 741,551,222 and Rhesus Macaque contacts totalled 351,837,113.

9. For Figure 3C and related analyses and conclusions, all the tissues shown have reduced expression in European populations which leads the authors to conclude about selection in the tissues most affected by anthrax. However, to make the claim that this selection is tissue-specific, a negative control must be included. This could be a tissue anthrax does not act on and an

effective control would be no difference in expression. As presented and without better justification the claims of tissue-specific adaptation are unsupported and should be removed and replaced with species- or population-specific adaptations.

Response: In the revised manuscript, we expanded our analysis of *ANTXR2* expression differences to all tissues analyzed by GTEx. As the reviewer indicated may be the case, we found lower *ANTXR2* expression in European relative to African populations in nearly all tissues with *ANTXR2* expression. We have revised the results and discussion section to indicate that decreased *ANTXR2* expression in Europeans appears to affect all tissues. Changes in the manuscript are too extensive to note in full here, but include a revised paragraph in the Results section (pp. 7):

Patterns of human variation in *ANTXR2* expression across tissues

To evaluate expression between different human populations, we divided GTEx RNA-seq data⁶⁰ into individuals with evidence of European and African ancestry based on the mitochondrial haplotype⁶¹. We found that individuals of African ancestry had higher *ANTXR2* expression, on average, compared with individuals of European ancestry, for the majority of tissues (n = 22 / 27) (Fig. 3D). Despite limited power to identify differences with relatively few samples that have evidence of African ancestry in GTEx, the change in muscle expression was statistically significant on its own (individuals of European ancestry/individuals of African ancestry log₂ fold change of 0.52, FDR=0.032 based on DESeq2). Additionally, combining p-values across tissues using Fisher's method also supported a significantly decreased *ANTXR2* expression in individuals of European ancestry in tissues overall (p = 0.00015, Fisher's method). Thus, our analysis identified population level differences in *ANTXR2* expression affecting the majority of tissues, with a lower expression in Europeans as would be expected based on the likely direction of positive selection due to interactions with anthrax disease.

10. *ANTXR2* and DICE RNA-seq analyses were performed by mapping macaque and human RNA-seq to a human mRNA using Salmon and default mapping parameters, and consequently those mappings used to compare expression levels between the species. Given the divergence between species at the level of DNA, validation that mapping RNA-seq to another species transcript is not biasing the results should be provided.

Response: We performed a reciprocal expression analysis in the revised manuscript, in which Salmon was used to map reads to both the human and rhesus macaque genomes (and additionally between human and baboon for new analyses added during the revision). We mapped RNA-seq data collected from both species to each genome and compared the ratio of *ANTXR2* expression between species. This validation analysis is presented in the revised **Figure S4**. We find similar ratios between the reciprocal analyses (H/R=0.16 mapped to hg19, H/R=0.14 mapped to rhemac10) and (H/B=0.21 mapped to hg19, H/B=0.10 mapped to papanu4), indicating that mapping to hg19 may be a conservative estimate of the true difference in expression. The Methods section has been updated to reflect this additional control (pp. 17):

Salmon reciprocal comparison

Reciprocal salmon analyses were performed by mapping to both human/rhesus macaque genomes and both human/baboon genomes for a subset of samples to test the validity of performing Salmon on non-human primate data using the human genome. Both species were mapped to each genome and the ratios were

compared between species for expression (see Fig. S4). For mapping to hg19, transcript NM_001145794.1 was used for all species. Transcript XM_015138456.2 was used to map to rheMac10 and transcript XM_021938471.1 was used to map to panAnu4. There are similar ratios between the reciprocal analyses (H/R=0.16 mapped to hg19, H/R=0.14 mapped to rheMac10) and (H/B=0.21 mapped to hg19, H/B=0.10 mapped to panAnu4) that suggest mapping to hg19 may be a conservative estimate of the true difference in expression. Thus, we were comfortable with using hg19 for all Salmon analyses.

Additionally, we have also performed an analysis using each species reference coordinate system using our own RNA-seq data. In this analysis, we mapped reads to each species reference genome using STAR and then used CrossMap to convert reads to an hg19 coordinate system (as described above). We note that since this additional analysis involves a full alignment using STAR, it is too time consuming to conduct on the full population-scale data collected from DICE and other public data sources. Nevertheless, when run on our own RNA-seq data, this alternative mapping strategy provides exactly the same conclusion as the Salmon analysis, helping to further raise our confidence that results were not driven by mapping biases. These results are shown in **Supplementary Fig. 5**, and described in the revised Methods section.

Minor comments:

- It would be helpful to have line numbering for revisions.

Response: We have added line numbers to the revised manuscript.

- Pg. 3: malaria should not be capitalised

Response: This has been corrected.

- Pg. 3: It would be helpful to explain which period is “3-6ka”.

Response: We have updated the shorthand in the text to better explain what time period we were referring to (pp. 1).

B. anthracis then radiated from Africa into the middle east and Europe approximately 3000-6000 years ago in the mid-Holocene, possibly following the spread of Neolithic agricultural practices¹⁶.

- Which version of NCBI RefSeq was used for the annotation? The most current version includes a processed pseudogene in human between ANTXR2 and PRDM8 which might affect both the transcription seen and definitions of “gene deserts”

Response: We also agree that we should be careful with our descriptive language, and we understand that the term gene desert has a very specific meaning. Out of an abundance of caution, we have removed the term “gene desert” from the text.

The RefSeq version used is 105.20190906, which does not include the pseudogene. To provide a full picture of the locus to readers, we have included the pseudogene (from GENCODE v36) in a new Supplementary Figure (**S20**). We agree that the divergent transcription unit upstream of the PRDM8 promoter may reflect transcription of this processed pseudogene.

- Figure 1A, S11B – what do the track colors represent? In PRO-seq tracks blue seems to denote the negative strand, but the same color is also used in the dREG track in seemingly positive direction.

Response: We have updated the colors of the dREG tracks to clarify things and have noted the color key in the figure legends.

Figure 2. Changes in *ANTXR2* cis-regulatory element activity and chromatin structure. A) Genome browser shot of CD4+ T cell PRO-seq (normalized by reads per million, plus strand in red, minus strand in blue) in human, chimpanzee, and rhesus macaque and regulatory elements and regulatory elements predicted by dREG (teal).

- Figure 2C and S7 – what is shown on the y axis?

Response: We have updated both figures to show the number of contacts for each anchored locus (virtual 4C-seq).

Figure 2. Changes in *ANTXR2* cis-regulatory element activity and chromatin structure. C) Virtual 4C-seq signal calculated based on the number of Hi-C contacts for CREs tested in the luciferase assay. CREs with decreased activity in human are colored. The anchor symbol at each CRE represents the bait region that all contacts are derived from. The arrow at the end of the arc denotes the *ANTXR2* promoter.

- Figure 2B, S6 – what are the units on the color scale? It would also be helpful to have the genes and CREs annotated here.

Response: We have updated the color scale to provide units in terms of normalized contacts, similar to the Virtual 4C-seq figures. We have also included some gene and CRE annotations to provide the reader with additional context.

- Pg. 6 - “majority of CREs in the gene desert”. Specific numbers should be provided.

Response: We have removed the vague language and included the specific number of CREs instead (pp. 4).

The 12 candidate CREs were spread throughout the *ANTXR2* locus, including inside of the *ANTXR2* transcription unit, situated between *ANTXR2* and the neighboring protein-coding gene, *PRDM8*, and within the region surrounding the *PRDM8* transcription unit (Fig. 2A).

- Figure 1C,S7 – what are the arrows and anchor? This information should be provided in the legend or figure.

Response: We have updated the figure legend to better explain the virtual 4C-seq plots. The revised legend reads as follows (pp.28-29):

Figure 2. Changes in *ANTXR2* cis-regulatory element activity and chromatin structure. C) Virtual 4C-seq signal calculated based on the number of Hi-C contacts for CREs tested in the luciferase assay. CREs are colored. The anchor symbol at each CRE represents the bait region that all contacts are derived from. The arrow at the end of the arc denotes the *ANTXR2* promoter.

- Figure S8 – what is the legend referring to (namely: TFH,TH17,TH1)?

Response: We have updated the figure legend to better explain the eQTL supplementary figure. The revised figure legend reads as follows (now S13 instead of S8):

Supplementary Figure 13. *ANTXR2* eQTLs in humans. A) Top: eQTL effect size (calculated from the DICE database) in different T-cell populations (T follicular helper cells (TFH, blue), T helper cells 17 (TH17, orange), and T helper cells type 1 (TH1, green)). Bottom: Pairwise LD correlation heatmap between SNPs around the *ANTXR2* locus. Expression eQTLs for *ANTXR2* fall into two main regions: within the gene and upstream of *ANTXR2* around *PRDM8*. Genic eQTLs are in linkage disequilibrium (LD) and have a positive effect on *ANTXR2* expression. Upstream eQTLs fall within two blocks of LD and have a negative effect on *ANTXR2* expression.

- Pg. 6 “the sweep had minimal overlap” – how is “minimal” defined? What is the expectation? The sweep overlaps 2/9 CREs.

Response: We have updated the language in the text to be less subjective and stated that there was overlap with 2 CREs (pp. 6).

This sweep overlapped 2 CREs associated with decreased *ANTXR2* expression between humans and non-human primates (CRE2 and CRE3).

- Figure 4B and S11 – what does “Net Synteny” represent?

Response: Net synteny tracks show the position of regions that have one-to-one orthologs in the chimpanzee and rhesus macaque genomes. We have updated the figure legends of these two figures to include this information (pp. 32).

Figure 4. Genetic differentiation in European populations consistent with multiple phases of selection driving changes in *ANTXR2*. B) PRO-seq (normalized by reads per million) and dREG signal (pictured in teal) at CRE2. rs41407844 falls within a dREG peak shared by chimpanzee and rhesus macaque and the ancestral allele G is conserved within the primate lineage. Conservation of CRE2 between humans and non-human primate tracks show human-specific SNPs, human/chimpanzee differences (SNPs in black, INDELS in red), human/non-human primate differences (SNPs in black, INDELS in red), and 100-way conservation. Net synteny tracks show the position of regions that have one-to-one orthologs in the chimpanzee and rhesus macaque genomes.

- Pg. 16 – the end of “Mapping orthologs between species” – is this section missing a word of a method/software at the end?

Response: We have updated this section with the missing software at the end of the sentence (pp. 16).

Mapping orthologs between species

Cross-species comparison of genomic coordinates and genes was based on the methods in Danko et al. 2018³⁵. Briefly, all datasets for chimpanzee and rhesus macaque were converted to the human assembly (hg19) using CrossMap⁵⁸. Reciprocal-best (rbest) nets were used to convert genomic coordinates between genome assemblies using the chaining and netting technique from Kent et al. 2003⁵⁹. We downloaded rbest nets for hg19-panTro4, hg19-rheMac10 from the UCSC Genome Browser.

Reviewer #2:

Choate et al., hypothesized that human may had adapted to anthrax disease during several period of human evolution. To test that hypothesis they combined population genetic tools with functional data to look for evidence for selection in the *ANTXR2* gene locus. Their results suggest that there has been multiples waves of positive selection in *ANTXR2*, which are associated with lower expression of the gene. The paper is well written, the results are exciting and novel, and most conclusions are well supported by the data presented. My main critique is that often cases it feels that the data used was based on availability rather than something specifically designed to address the questions being posed. This does not necessarily diminish my overall enthusiasm for the study but it raises a number of questions that I think should be addressed before publication.

Response: We are grateful to the reviewer for their interest and constructive comments throughout the review! We have made our best effort to address each of the comments, and present our detailed responses below.

Specific comments

1. The first section of the Results is focused on the comparison of *ANTXR2* expression levels in CD4 T cells between humans and Rhesus macaques. From a biological perspective, is there a reason to focus on CD4 T cells? Some rational should be provided.

Response: We have clarified the rationale for our initial focus on CD4+ T-cells in the revised manuscript.

Briefly, CD4+ T-cells have a key role in initiating adaptive immune responses. CD4+ T-cells circulate throughout the body and evaluate whether foreign material is present. When a possible infection is detected, CD4+ T-cells initiate an adaptive immune response by secreting signaling peptides called cytokines, which attract a wide variety of immune cells to the site of infection. Thus, CD4+ T-cells are frequently located in close proximity to pathogens and have an organismal role in initiating immune responses. In the specific context of anthrax disease, previous literature indicates that *B. anthracis* evades adaptive immune responses by dampening cytokine production in CD4+ T-cells. This could prevent infected individuals from launching a broader adaptive immune response that could clear the infection.

We have clarified the rationale for examining CD4+ T-cells at the beginning of the revised Results section (pp. 3):

Changes in *ANTXR2* transcription and mRNA in early human ancestors

Anthrax infection produces toxins that target immune cells to dampen the immune response and aid in spore migration throughout the body²¹. A key component of the adaptive immune system, CD4+ T-cells, have measurable changes in activation, cytokine production, chemotaxis, and differentiation after anthrax infection^{22–24}.

In addition, the author talks about differences in *ANTXR2* expression between humans and non-human primates but only data for Rhesus macaques are shown. The fact that they see higher expression levels in Rhesus macaques as compared to humans cannot be interpreted as representative of most non-human primates. The author should at the very least include the expression levels of *ANTXR2* in Chimpanzee for which expression data in CD4 T cells is also available.

Response: We agree that we need to be careful attributing expression in rhesus macaque with all non-human primates. In the revised manuscript, we added new RNA-seq data from baboon CD4+ T-cells, which have similar expression to rhesus macaque. In addition, **Figure 1C** shows *ANTXR2* expression (estimated using PRO-seq data) from CD4+ T cells in human, chimpanzee and rhesus macaque. All of the data collectively support a model in which humans lost *ANTXR2* expression relative to a common ancestor with chimpanzee, based on maximum parsimony assumptions. In the revised manuscript, we state which non-human primate species are used in each comparison more clearly. In addition, we also focus on the human loss (which is the most important point here), and more clearly state the maximum parsimony assumption on which we base our conclusions. See especially (Results, pp. 3):

Changes in *ANTXR2* transcription and mRNA in early human ancestors

Consistent with our hypothesis, *ANTXR2* was transcribed at 8-fold lower levels in human CD4+ T-cells compared to non-human primates at the level of both Pol II loading (chimpanzee and rhesus macaque) and mRNA (rhesus macaque and baboon) ($p < 0.0001$) (**Fig. 1B-C**; **Fig. S2A-B**; **Fig. S3**). Expanding our analysis of RNA-seq data from CD4+ T-cells to 91 humans confirmed the loss of *ANTXR2* expression in all of the available human data²⁸ (**Fig. 1D**, **Fig. S4**). Under the assumption of maximum parsimony, our comparative analysis indicates that *ANTXR2* was down-regulated in the human lineage.

2. In Figure 1D the authors compare expression levels of *ANTXR2* between a panel of 85 humans and two rhesus. It would be extremely informative to expand the analyses to include additional rhesus samples and immune cell types. For example, the authors could use the data from the paper by Snyder-Mackler et al. Science (PMID: 27885030) for which data on 5 immune cell types from a panel of 45 Rhesus macaques is available. Knowing in what immune cell types we see a difference in *ANTXR2* gene expression levels would help inform on the putative mechanisms that have been selected for.

Response: As suggested by the reviewer, we have included matched RNA-seq data from our lab for CD14+ monocytes in human, rhesus macaque, and baboon (**Fig. S5**). Additionally, we have included a comparison between unmatched data for CD4+, CD14+, and other immune cell types between human data (from DICE) and rhesus macaque data (from Snyder-Mackler et al.) (**Fig. S6**). Collectively, the results of our new analysis show that humans have lower *ANTXR2* in expression in CD4+ T-cells, CD8+ T cells, B cells, and monocytes. We found no significant differences between NK cells, although this lack of a signal may result from a much lower level of expression in all species in NK cells. Collectively, thanks to this fantastic suggestion from the reviewer, we now show that *ANTXR2* expression differences are present in both lymphoid and myeloid lineages, and affect most cell types in the immune system. The revised analysis is described in detail in the Results section, pp. 3:

Changes in *ANTXR2* transcription and mRNA in early human ancestors

Expanding our analysis of RNA-seq data from CD4+ T-cells to 91 humans confirmed the loss of *ANTXR2* expression in all of the available human data²⁸ (**Fig. 1D, Fig. S4**). Under the assumption of maximum parsimony, our comparative analysis indicates that *ANTXR2* was down-regulated in the human lineage.

We examined whether expression changes in *ANTXR2* were specific to CD4+ T-cells. We found that *ANTXR2* was expressed at significantly lower levels in CD14+ monocytes isolated from humans than from rhesus macaque or baboon ($p < 0.05$) (**Fig. S5**). Expanding this analysis using published RNA-seq from multiple immune cell types in humans²⁸ and rhesus macaque²⁹ identified ~10-fold lower *ANTXR2* expression in the majority of human immune cell types, including B cells, CD8+ T-cells, and monocytes (**Fig. S6**). Differences in *ANTXR2* expression between species were not identified in natural killer cells, although this cell type had relatively low levels of *ANTXR2* mRNA in both species. Next, we asked whether changes in *ANTXR2* expression were specific to the immune system using microarray data³⁰ comparing three tissues (heart, kidney, and liver) in human, chimpanzee, and rhesus macaque. We found no consistent human-specific decrease in *ANTXR2* expression in these other tissues, suggesting that decreased expression of *ANTXR2* in humans is restricted to the immune system (**Fig. S7**). Collectively, analysis of *ANTXR2* expression data was consistent with our hypothesis of ancient evolutionary changes affecting *ANTXR2* expression in the immune system during the early divergence of humans from other primates.

3. The authors provide several lines of evidence that the genetic diversity patterns around the *ANTXR2* locus are not compatible with neutral evolution, particularly among European populations. To my surprise, however, the authors did not try to connect the signals of selection observed to the eQTL for *ANTXR2* present in the region. The expectation is that the allele(s)/haplotype(s) associated with lower gene expression would be the ones showing increased allele frequency in European populations. Is that the case? Any evidence for selection for these eQTL based on haplotype-based tests like iHS?

Response: We agree that it would be outstanding to have a clear example in which an eQTL was also under selection by an iHS type summary statistic. During the revisions, we looked through published iHS data in human populations. Briefly, we analyzed the region around *ANTXR2* using the human genome iHS test done by Voight et al. PLOS Biology 2006. Of 104 eQTLs (or SNPs in close LD) identified in the DICE dataset, 11 were analyzed in the iHS scan. We found a block of high-iHS signal around the *ANTXR2* promoter region. This block is not significant by the Voight et al. definition alone, but was in the 94.5th quantile of genome-wide iHS distribution (see **Figure below**). This high quantile value may suggest that the locus swept an ancestral allele which is consistent with our model. However, as this locus did not pass the stringent cutoff threshold selected by Voight et. al., we do not think it is helpful to include in the final manuscript. Thus, while we have found a few promising leads that may warrant future investigation in larger datasets, we have not identified a definitive result that we feel warrants presentation in the main text.

Figure: Genome-wide iHS scores

4. The authors talk about the fact that selection might have acted on regulatory elements that are specific to tissues like heart, lung and liver. There has been several comparative studies of gene expression in these tissues between human, rhesus and chimps (e.g., Blekhman et al, PLoS Genetics, PMID: 19023414, among several others). It would be important to show that in these tissues *ANTXR2* shows lower expression levels in humans as compared to other primates (the expectation based on their proposed evolutionary model).

Response: We analysed the microarray data from Blekhman et al. for heart, kidney, and liver in human, chimpanzee, and rhesus macaque. We did not observe the same signal for a human-specific decrease in *ANTXR2* expression in the heart, kidney or liver that we observed in immune cells. We think this result indicates that changes in *ANTXR2* expression early during human evolution primarily affect the immune system, whereas later changes that primarily affect

Europeans populations affect a larger group of tissues. We present these new analyses in the Results section. See especially the sections on ‘*Changes in ANTXR2 transcription and mRNA in early human ancestors*’ section (pp. 3) and the ‘*Selective sweep affects ANTXR2 expression in multiple tissues central to anthrax disease pathogenesis*’ section (pp. 6). Relevant text is also shown here:

Changes in ANTXR2 transcription and mRNA in early human ancestors

Next, we asked whether changes in *ANTXR2* expression were specific to the immune system using microarray data³⁰ comparing three tissues (heart, kidney, and liver) in human, chimpanzee, and rhesus macaque. We found no consistent human-specific decrease in *ANTXR2* expression in these other tissues, suggesting that decreased expression of *ANTXR2* in humans is restricted to the immune system (Fig. S7).

Selective sweep affects ANTXR2 expression in multiple tissues central to anthrax disease pathogenesis

A recent hard sweep discovered by SweepFinder2 may affect *ANTXR2* expression in different ways than observed for the ancestral expression changes between human and non-human primates. Comparative analysis of RNA-seq and microarray data between human and non-human primates (presented above) indicated that changes in *ANTXR2* expression were primarily limited to the immune system (Fig. 1; S6; S7).

Reviewer #3:

This paper presents a novel, thorough and very compelling investigation of the function and evolution of anthrax susceptibility in humans, but I have concerns about the later part of the paper, which focuses on within-human evolution. The results are not sufficient to support the conclusion of a significant difference in Europeans, let alone to conclude that natural selection, rather than random drift or other population events, is responsible. This is, of course, a particularly fraught topic given that it involves comparing Europeans to Africans and Asians, and needs to be handled with great care.

Overall, I really enjoyed reading this manuscript, and for the first half have minor only comments. I was incredibly impressed by the functional investigation of the *ANTXR2* locus, which really took the genomic story of divergence between species and figured out what was happening in terms of genome function. The authors' choice to start with the functional analysis at this locus was very effective, and drew me into the science.

Response: We are grateful to the reviewer for their thoughtful and constructive review of our manuscript. We appreciate the constructive comments, which we have made our best effort to fully address in our revision. We particularly wish to emphasize here that it was not our intention to wade into a delicate area where our meaning could be misconstrued. We have made our best effort to address this, and other, comments raised by the reviewer. We are, of course, happy to make additional changes that would make the reviewer more comfortable with our presentation.

In the results section, it gets a bit confusing to follow which comparisons are happening where, and adding a bit more specificity would help, including more details in the figures. For example, in the first paragraph of the results, the authors refer to comparing human CD4+ T-cells to 85 humans, but some information on the ancestry of those 85 humans seems relevant given the hypothesis of evolution in the paper.

Response: We have added additional detail to figure legends in all of the figures in the paper, including information on the ancestry of the DICE dataset. This DICE database contains predominantly individuals that self-report as white or Asian (n=45 and 20, respectively).

Below is the distribution of the ethnicities as determined by GRAF (<https://www.ncbi.nlm.nih.gov/projects/gap/population/cgi-bin/StudySubjectAncestry.cgi?selRaces=0%2C1%2C2%2C3%2C4%2C5&isBeyond=&phs=1703&version=2&minSnps=10&dotSize=2&rotx=0&showCompPop=&showCut=&racePick=0&racePick=1&racePick=2&racePick=3&racePick=4&racePick=5&graphWidth=600&dotSide=2&minPopSnps=10&rotatex=0&xmin=1.2&xmax=2&ymin2=1&ymin2=1.8&popType=1>). We have included information about this in the figure caption and in the Methods section. Of note, we also pulled data from the remaining 6 individuals from the dataset so now n=91.

I'm also unclear which comparisons were significant - the authors refer to the 8-fold downregulation in humans compared to non-human primates, but there are no measures of statistical significance in either Figure 1C or Fig S2B.

Response: We have added measures of statistical significance to the comparisons in **Fig. 1C** and **Fig. S2B** in the revised manuscript. In all cases, p-values using an appropriate statistical test are now presented in the figure legends. All of the changes between humans and non-human primates were highly unlikely to happen by chance ($p < 0.0001$, DESeq2).

In contrast, figure S3 has a statistical analysis, but doesn't highlight what appears to be a much more substantial difference between both human populations and rhesus macaque than between the human populations, or give a measure of magnitude - it just states that all the differences are significant.

Response: We agree with the reviewer - there are substantial and statistically significant differences between human populations in the comparison illustrated in **Fig. S8**. This figure highlights differences between human populations, which we focus on in the 2nd half of the paper. In the revised manuscript, we now refer back to **Fig. S8** when making the points about population variation. See, for example, changes to the results section, including (pp. 3-4):

Changes in *ANTXR2* transcription and mRNA in early human ancestors

To more precisely date the occurrence of expression changes, we analyzed RNA-seq data from peripheral blood mononuclear cells (PBMC), a cell population largely composed of immune cells, in multiple human populations. We found that *ANTXR2* expression was lower in two African populations, including Batwa, a hunter gatherer population historically in the Great Lakes region of Africa, and Bakiga, agriculturalists from neighboring Rwanda and Uganda, compared to rhesus macaque^{31,32} (**Fig. S8**). We also noted substantial variability within human populations, including a higher *ANTXR2* expression in hunter-gatherer than agrarian populations (examined in detail, below). These results suggest that expression differences between human and rhesus macaque began before human populations expanded and migrated out of Africa.

In the second paragraph of results, the authors switch from CD4+ t-cells, which they had justified using as a model, to K562 cells, but don't provide any description/justification for using that cell line in the main text.

Response: We have clarified the rationale for using K562 cells in CRISPRa experiments in the main text. Briefly, while it would be preferable to work directly with primary CD4+ T-cells, primary T-cells are extremely challenging to genetically manipulate in ways that are required for CRISPRa. We therefore switched to K562, a hematopoietic cell type that is highly genetically tractable. We have clarified the rationale for this choice in the main text and methods, see specifically (pp. 4 and 12):

Changes in *ANTXR2* transcription and mRNA in early human ancestors

We selected K562 cells as a model system because K562 is a human hematopoietic cell line that could be manipulated in culture and showed a basal level of *ANTXR2* expression.

Methods

CRISPRa in K562

Use of K562 cell line— K562 was used for CRISPR experiments because of its ability to be transfected and manipulated in culture. K562 possesses basal expression of *ANTXR2*, allowing for comparison to CRISPR activation. A key component of the CRISPRa experiment is treating CRISPR manipulated cells with anthrax toxins to measure the effect of expression on viability. K562 cells remained viable and healthy enough after transfection to be used in toxin viability assays. Cell lines that more closely replicate the trans environment of human CD4+ T-cells, such as Jurkat, were not viable for days after transfection.

In figure 1E and 1F (despite using much more informative box plots in the supplemental figures) the authors present bar charts, and don't show either the individual data points or even give details on the number of samples. It's not shown if any of the comparisons in 1E are significant.

Response: We have added individual points to our revised **Figure 1E** and **1F**. In addition, we have also included statistical comparisons in the text and figure legends. See especially (pp. 4 and 27):

Figure 1 E) CRISPRa induction in K562 cells results in significantly increased *ANTXR2* expression after 24 and 48 hours ($p < 0.01$, t-test) measured using qRT-PCR.

In the multispecies comparison (Figure 2), it's not clear whether the authors looked for regulatory elements present in the primates and missing in the humans, or just at elements conserved in humans.

Response: We examined regulatory elements for which we could identify a signature of regulatory activity using genomic data from human, chimpanzee, and rhesus macaque. Technically, our analysis used PRO-seq data from all three primate species for signatures of transcription initiation that are characteristic of enhancer and promoter regions using the dREG software package (Wang et. al. (2018) Genome Research). We have clarified that our analysis examined data from all three primate species in the Results section. See especially (pp. 4):

Independent DNA sequence differences distributed in multiple CREs underlie *ANTXR2* expression changes

We asked whether cis-regulatory changes could account for transcriptional divergence in *ANTXR2* between species. We set out to identify cis-regulatory elements (CREs) near the *ANTXR2* locus in primates. We used 12 transcription initiation regions previously identified in CD4+ T-cells from human, chimpanzee, and rhesus macaque using dREG^{35,36}, a tool which identifies regulatory elements using patterns of nascent transcription in PRO-seq data (see Methods).

In figure 2A, I don't know what the red and green vertical shading means. Is it somehow human and macaque like is shown in the key in panel 2C?

Response: We have made several changes to Figure 2 with the aim to make it easier to understand. In the revised manuscript, shading is used to denote candidate CREs that have a significant change in luciferase signal between human and at least one of the non-human primate

species (chimpanzee or rhesus macaque). This is now stated more clearly in the revised legend of Figure 2A (pp. 28):

Figure 2. Changes in *ANTXR2* cis-regulatory element activity and chromatin structure. A) Genome browser shot of CD4+ T-cell PRO-seq (normalized by reads per million, plus strand in red, minus strand in blue) in human, chimpanzee, and rhesus macaque. Regulatory elements predicted by dREG are shown in teal. CREs tested in this study (CRE1-9) are shown below the data tracks. CREs having decreased activity in human (as determined by luciferase in 2D) are shaded. Overlap with validated CRISPR targets⁴⁶ that result in reduced *ANTXR2* expression are shown below the CREs in purple.

In Figure 2C, there is no explanation of what the y axis is. I'm not sure how that panel demonstrates "the main focal contacts observed in human Hi-C data had a higher contact frequency ... ", or if there is a difference between the three CRE. In figure 2D, the authors chose to use a bar plot rather than a boxplot (maybe this is standard - I'm not sure) and didn't show sample points or give sample numbers. In 2D, the non-significant comparisons are not shown, which bothers me - I'd rather see all the results.

Response: We have performed all of the changes suggested by the reviewer. We have added additional axes labels to the Virtual 4C-seq plot in **Figure 2C**. We added individual data points to the barplots in **Figure 2D** and now show the number of samples in the figure caption. Additionally, we now include all luciferase barplots in the main text, instead of splitting them between the main text and supplement based on the statistical significance of differences between humans and either non-human primate. Finally, to clarify how CREs are arranged in relation to each other, we have added representations of non-significant CREs into **Figure 2A** and **Figure 2C**. Changes to the text in response to this comment are too extensive to enumerate exhaustively here, but please see (among other places), changes to figure 2 and its associated figure legend (pp. 29):

Figure 2. Changes in *ANTXR2* cis-regulatory element activity and chromatin structure. C) Virtual 4C-seq signal calculated based on the number of Hi-C contacts for CREs tested in the luciferase assay. CREs with decreased activity in human are colored. The anchor symbol at each CRE represents the bait region that all contacts are derived from. The arrow at the end of the arc denotes the *ANTXR2* promoter.

In 2D, I'm not sure why Jurkat cells are now being used, and not T-cells or K562 cells. I'm very sure there is a good reason, but the manuscript would be more readable with a bit more context scientists less familiar with this type of assay.

Response: We have clarified the rationale for using Jurkat T-cells, instead of K562 or primary CD4+ T-cells, as a model in the luciferase assays. The primary rationale for switching to Jurkat is that for luciferase assays to provide insight into primary CD4+ T-cells, it is critically important to match the trans-environment in a cell type that can be transfected with high efficiency. As noted above, primary CD4+ T-cells are challenging to transfect, preventing their use in this experiment. Jurkat T-cells are a CD4+ cell line that is more similar to a primary CD4+ T-cell than K562 cells. In particular, to justify the switch to Jurkat T-cells, we now show that similar CREs are active in both Jurkat T-cells and in primary CD4+ T-cells, whereas K562 is distinct (see Supplementary Fig. 14). We note that the trans-environment is much more important in luciferase assays

performed here than in the *ANTXR2* overexpression experiments, described above. We have clarified the rationale for using these different cell lines in the results and methods sections. See in particular (pp. 4-5 and 12):

Changes in *ANTXR2* transcription and mRNA in early human ancestors

We selected K562 cells as a model system because K562 is a human hematopoietic cell line that could be manipulated in culture and showed a basal level of *ANTXR2* expression.

Independent DNA sequence differences distributed in multiple CREs underlie *ANTXR2* expression changes

We used Jurkat human leukemic CD4+ T-cells as a model trans-environment which recapitulates the pattern of transcription in the *ANTXR2* locus observed in primary T-cells better than alternative models, including K562 (Fig. S14).

Methods

Use of K562 cell line— K562 was used for CRISPR experiments because of its ability to be transfected and manipulated in culture. K562 possesses a basal expression of *ANTXR2*, allowing for comparison to CRISPR activation. A key component of the CRISPRa experiment is treating CRISPR manipulated cells with anthrax toxins to measure the effect of expression on viability. K562 cells remained viable and healthy enough after transfection to be used in toxin viability assays. Cell lines that more closely replicate the trans environment of human CD4+ T-cells, such as Jurkat, were not viable for days after transfection.

The statements about expected conservation and divergence of CREs feels very vague. Is there any way to measure how significant the “significant degree of cis-regulatory divergence” between human and non-human primates. is? E.g. how often would such a pattern of conservation in macaque/chimpanzee & divergence in humans be expected in the genome, given what we know of CRE conservation.

Response: We have revised the main text and figures to be more quantitative and precise about the divergence between human and non-human primates at the *ANTXR2* locus. Briefly, several papers have shown that it is relatively rare for different species to show major, coordinated changes in the activity of candidate CREs within each locus (see especially: Prescott et. al. (2015) Cell; Danko et. al. (2018) Nature Eco Evo). Moreover, the rare exceptions are associated with changes in transcription of the nearby genes. Motivated by these observations, we examined whether the transcriptional activity of regulatory elements in the *ANTXR2* locus had coordinated changes in a consistent direction. We examined this question in two semi-independent ways: First, we examined the quantitative differences in transcription within all candidate CREs in the locus between humans and non-human primates. This analysis revealed that candidate CREs decreased expression in the human lineage by a median of 2.3-fold, which represents a large and coordinated change in CREs across the locus. Second, we examined whether these differences in transcription were significantly different relative to the genome as a whole. In this analysis, we examined the genome-wide statistical significance of changes in transcription near each candidate CRE. This genome-wide analysis revealed that at least three of the candidate CREs were significantly changed in the human lineage, compared with chimpanzee and rhesus macaque: all down-regulated in human. Taken together, these analyses provide quantitative support for uncharacteristic human-specific changes in transcription at the *ANTXR2* locus. This

combination of new analysis and clarifications is presented in **Supplementary Fig. S10** and highlighted in the main text (pp. 4):

Independent DNA sequence differences distributed in multiple CREs underlie *ANTXR2* expression changes

Different species rarely show major, coordinated changes in the activity of candidate CREs within a locus, and the exceptions are associated with major changes in gene expression and the appearance of new traits^{37,38}. Testing each putative CRE for differential transcription, a hallmark of regulatory activity at enhancers³⁹, revealed a strong bias for decreased transcription in the human lineage across the *ANTXR2* locus (median 2.3-fold lower transcription in human; $p = 0.001$; Wilcoxon rank sum test; **Fig. S10**). At least three CREs were differentially transcribed on the human lineage using a conservative test for differential transcription (DESeq2⁴⁰ FDR corrected p -value < 0.05), all of which were down-regulated. The significant degree of cis-regulatory divergence between human and non-human primates at multiple annotated CREs is uncharacteristic of randomly selected genomic regions. Moreover, multiple changes at distinct CREs suggests that cis-acting loci, rather than trans-acting factors, underlie much of the divergence in *ANTXR2* expression.

Minor points In 3rd paragraph, “dReg” is used without any explanation of what that is. The word “notably” is used, twice. Particularly in the first usage, there isn’t any context to understand why the following observation is notable.

Response: We provided a better explanation of the dREG tool in the main text (pp. 4) and Methods section (pp. 17) and removed occurrences of the word “notably”, which we agree was confusing.

Independent DNA sequence differences distributed in multiple CREs underlie *ANTXR2* expression changes

We asked whether cis-regulatory changes could account for transcriptional divergence in *ANTXR2* between species. We set out to identify cis-regulatory elements (CREs) near the *ANTXR2* locus in primates. We used 12 transcription initiation regions previously identified in CD4+ T-cells from human, chimpanzee, and rhesus macaque using dREG^{35,36}, a tool which identifies regulatory elements using patterns of nascent transcription in PRO-seq data (see Methods). A comparison with DNase-I hypersensitivity and H3K27ac ChIP-seq data in human CD4+ T-cells did not identify additional candidate CREs in this locus (**Fig. S9**), indicating that dREG was sensitive enough to pick out the most important candidate CREs.

Methods

Predicting CREs using dREG

We previously used dREG to identify candidate CREs using PRO-seq data from human, chimpanzee, and rhesus macaque CD4+ T-cells³⁸. dREG uses a machine learning strategy called support vector regression to score genomic sites based on patterns of PRO-seq data learned from transcription start sites in a different cell type^{35,36}. In Danko et al. 2018³⁸, scores were computed on the combined signal for all individuals to maximize statistical power. dREG scores in chimpanzee and rhesus macaque were converted to hg19 coordinates using CrossMap and their corresponding reciprocal-best nets. The false discovery rate was set to be $< 7\%$, which corresponds to a dREG score threshold of 0.3. Here we used dREG sites near *ANTXR2* from our Danko et al. 2018. Because sites of divergently initiating Pol II identified by dREG, called transcription initiation regions, frequently act in concert with other nearby initiation regions^{75,76}, we merged candidate dREG regions within 1500 bp using BEDTOOLS. After merging all candidate CREs, we had 14 candidates within the *ANTXR2* locus for analysis.

My concerns start with the section on the recent selective sweep in Europeans. It seemed very odd to me that the authors presented their own, new, analysis limited to just one chromosome, using the 1000G project data. Selection scans should only be done in the context of the whole genome, in order to show how unusual a locus is, given all the patterns that other population events can leave in the genome.

Response: In the revised manuscript, we ran Sweepfinder2 (DeGiorgio et al. (2016) Bioinformatics) on the whole genome. In the revised **Figure 3A and Supplemental Fig. 16**, we present the new selection scan with updated CLR values. This new genome-wide analysis highlights the *ANTXR2* locus as an outlier, with CLR values at the *ANTXR2* locus in the top 0.5% percent (99.5 percentile) of scores genome-wide. This represents a genome-wide outlier, and is slightly more significant than reported based on chromosome 4 in the original paper top 2% (98th percentile). We have revised the main text to illustrate this scan. See especially (pp.6):

Recent selective sweep in Europeans upstream of *ANTXR2*

To determine how selection has influenced modern human populations at the *ANTXR2* locus, we computed the composite-likelihood-ratio (CLR) of a selective sweep genome-wide in four human populations (Europeans [CEU], East Asians [CHB and JPT], and Africans [YRI]) using SweepFinder2⁵⁵. Consistent with the hypothesis that anthrax disease in Europeans led to a relatively recent population specific adaptive response, we found a candidate selective sweep upstream of *ANTXR2* (**Fig. 3A**). The selective sweep had a higher CLR in European (CEU) than 99.5% of other loci genome-wide (**Fig. S16**).

Furthermore, many other groups have already done genomewide scans using exactly the same datasets used in this paper. The authors should show the results from those previous studies for the *ANTXR2* locus. If those studies did not conclude there was evidence for positive selection at this locus, why not?

Response: We have examined the literature doing genome-wide positive selection scans in humans. Kimura et. al. (2007), highlights the *ANTXR2* locus as a genome-wide outlier, stating that: “It is interesting that [...] anthrax toxin receptors [*ANTXR2*], show signatures of selective sweeps, which may indicate cases in human history of fights against specific infectious diseases.” Additionally, Raj et al. (2013) reports variants that are associated with inflammatory disease as part of a genome-wide scan for positive selection in Europeans and includes a SNP within *ANTXR2*. We now cite this paper as an additional source of support that *ANTXR2* is under positive selection in humans. In particular, see the Results section (pp. 6):

Recent selective sweep in Europeans upstream of *ANTXR2*

The *ANTXR2* locus and SNPs within *ANTXR2* have shown up in previous scans for positive selection in the human genome^{56,57}, which provides additional evidence that the locus surrounding *ANTXR2* is under positive selection.

The authors then look at *ANTXR2* gene expression. This paragraph is frustratingly vague. The information from Roadmap feels entirely descriptive. For example - “We found substantial numbers of DNase-I hypersensitive sites overlapping the interval of high CLR values in heart, lung, liver, and dermis”. I have no idea how interesting this really is. Across the genome, there

are lots of DNase-I hypersensitive sites in heart, lung, liver, and dermis, many of which will not be found in blood, and lots of intervals in the genome with high CLR values. The authors need to somehow demonstrate that there is something unusual about this locus.

Response: In response to this comment, we have expanded our analysis of both DNase-I hypersensitivity and *ANTXR2* mRNA abundance to all tissues analyzed by Roadmap Epigenomics and GTEx. Our new analysis shows that changes in *ANTXR2* expression between individuals of European and African descent appear to affect the majority of tissues in which *ANTXR2* is expressed. Many of these same tissues have evidence of DNase-I hypersensitive sites. Of these tissues, we note that muscle (including skeletal and cardiac), reproducibly (across multiple independent datasets) have the largest number of DHSs identified within the selected region.

In contrast, our analysis of existing comparative gene expression datasets (Blekhman et. al. PLoS Genetics (2008); which examined heart, liver, and kidney) indicates that changes between humans and non-human primates is tissue specific, primarily affecting *ANTXR2* mRNA in the immune system.

Taken together, these new analyses provide a more quantitative view of the DNase-I hypersensitivity in the region under selection, and reinforces the main thesis that *ANTXR2* expression has changed in at least two separate, independent phases during human evolution.

These new analyses are described at length in the main text, in the Results section. (see especially (pp. 7):

Patterns of human variation in *ANTXR2* expression across tissues

To evaluate expression between different human populations, we divided GTEx RNA-seq data⁶⁰ into individuals with evidence of European and African ancestry based on the mitochondrial haplotype⁶¹. We found that individuals of African ancestry had higher *ANTXR2* expression, on average, compared with individuals of European ancestry, for the majority of tissues (n = 22 / 27) (Fig. 3D). Despite limited power to identify differences with relatively few samples that have evidence of African ancestry in GTEx, the change in muscle expression was statistically significant on its own (individuals of European ancestry/individuals of African ancestry log₂ fold change of 0.52, FDR=0.032 based on DESeq2). Additionally, combining p-values across tissues using Fisher's method also supported a significantly decreased *ANTXR2* expression in individuals of European ancestry in tissues overall (p = 0.00015, Fisher's method). Thus, our analysis identified population level differences in *ANTXR2* expression affecting the majority of tissues, with a lower expression in Europeans as would be expected based on the likely direction of positive selection due to interactions with anthrax disease.

The evidence of increased expression in African individuals (figure 4C) is weak and problematic. First, the authors refer both to "African" (in the key) and "African American" (legend), which are obviously not interchangeable populations. Second, "African" is not usually used at all to refer to populations, given the massive diversity within that continent. If these individuals were in fact African American, and African refers simply to have some degree of African ancestry (which would be the case if based purely on mitochondrial genotypes) the range of African ancestry, and what this means for the analysis, should be specified.

Response: We thank the reviewer for bringing up such an important point. We have clarified that GTEx data population classifications based on mitochondrial haplotyping reflect individuals of African descent. We fully agree with the reviewer that these individuals will have a broad range of African and European ancestry. We have clarified this in the figure legends and main text, using the term individuals of African descent instead of Africans. We note that if this affects our analysis, it will likely underestimate the extent of differences in *ANTXR2* expression between individuals of different ancestry, as many individuals of African descent will carry the European allele at the *ANTXR2* locus due to admixture. See pp.7:

Patterns of human variation in *ANTXR2* expression across tissues

To evaluate expression between different human populations, we divided GTEx RNA-seq data⁶⁰ into individuals with evidence of European and African ancestry based on the mitochondrial haplotype⁶¹.

Finally, and most problematically, there is no significant difference in expression even on the single locus level, and no information on whether the pattern seen at this locus is unusual in the context of the whole genome.

Response: In the revised manuscript, we have expanded our analysis of *ANTXR2* expression to all tissues analyzed by GTEx. Despite the limited number of individuals of African descent in the GTEx dataset, several of the tissues had significantly decreased *ANTXR2* expression in individuals of European descent, including muscle (FDR < 0.03, DESeq2). Additionally, nearly all tissues had a decreased expression of *ANTXR2* in individuals of European descent. This decreased expression was highly significant based on a number of analyses, including Fisher's method where the combined p values showed significantly decreased *ANTXR2* in individuals of European descent overall (p=0.00015). Moreover, as noted above, using a dataset focused on individuals of African descent (which, as noted by the reviewer above, are likely admixed with individuals of European descent) likely underestimates the effect of decreased *ANTXR2* among Europeans. When we have data collected directly in different African populations (as for PBMCs shown in **Fig. S8**), we note a much larger difference between different human populations.

Patterns of human variation in *ANTXR2* expression across tissues

To evaluate expression between different human populations, we divided GTEx RNA-seq data⁶⁰ into individuals with evidence of European and African ancestry based on the mitochondrial haplotype⁶¹. We found that individuals of African ancestry had higher *ANTXR2* expression, on average, compared with individuals of European ancestry, for the majority of tissues (n = 22 / 27) (**Fig. 3D**). Despite limited power to identify differences with relatively few samples that have evidence of African ancestry in GTEx, the change in muscle expression was statistically significant on its own (individuals of European ancestry/individuals of African ancestry log2 fold change of 0.52, FDR=0.032 based on DESeq2). Additionally, combining p-values across tissues using Fisher's method also supported a significantly decreased *ANTXR2* expression in individuals of European ancestry in tissues overall (p = 0.00015, Fisher's method). Thus, our analysis identified population level differences in *ANTXR2* expression affecting the majority of tissues, with a lower expression in Europeans as would be expected based on the likely direction of positive selection due to interactions with anthrax disease.

The F_{ST} analysis is also unconvincing. F_{ST} is an incredibly noisy statistic, and I could not tell from the analysis whether any of the patterns observed at *ANTXR2* are anything more than random noise.

Response: To examine whether F_{ST} patterns near *ANTXR2* are random noise, we performed a new analysis to ask whether F_{ST} near the *ANTXR2* locus was a genome-wide outlier. We found that regions near *ANTXR2* are in the top 1% of all F_{ST} values genome-wide for comparisons between CEU (European) and either CHB, JPT, or YRI. We also examined a global F_{ST} metric that examines all three populations, and is therefore somewhat less noisy, and found similar results. These new analyses are shown in the revised **Figure 4D** and **4E** and the comparison to values across the genome is described in the main text (see pp. 7-8 and below).

We certainly agree with the reviewer that F_{ST} on its own, like any population genetic statistics, does not always provide a complete picture about natural selection. However, between the new genome-wide analysis of F_{ST} , the direct tests for positive selection using SweepFinder2, and the appearance of this locus in studies of selection by independent labs, we think there is substantial evidence that *ANTXR2* has recently been under selection in European populations.

Patterns of genetic differentiation supports continued human divergence upstream of *ANTXR2*

Finding a putative selective sweep specific to Europeans led us to ask whether the *ANTXR2* locus differentiates CEU from other human populations. To address this, we calculated relative population differentiation (F_{ST}) between Europeans (represented by CEU) and either East Asians (represented by CHB and JPT) or Africans (represented by YRI). The entire *ANTXR2* locus showed elevated differentiation between European and non-European populations, with a median value of 0.14-0.33 (64th-96th percentile genome-wide) (**Fig. 4A**). The greatest signal of differentiation occurred outside of the putative selective sweep, but overlapped upstream CREs (including CRE2) that diverged between human and non-human primates (**Fig. 4B**). One particularly interesting haplotype, directly adjacent to the region of high CLR, had a very high F_{ST} in Europeans relative to all other ethnic groups. This region included several SNPs overlapping CREs upstream of *ANTXR2*, including rs41407844—the allele frequency of which was correlated with reported variation in anthrax toxin sensitivity in lymphoblastoid cell lines⁵⁴: high frequency of the derived allele in Europeans (~0.85), intermediate in East Asians (~0.38), and low frequency in African populations (~0.17; **Fig. 4C-E**). This result, and the observation of elevated F_{ST} at CRE3, CRE4, and CRE9 in the absence of high CLR profile, suggests that Europeans have maintained genetic separation at loci derived from both incomplete soft sweeps on genetic variation from early human divergence and a hard sweep associated with recent European anthrax exposure (**Fig. S20**).

REVIEWERS' COMMENTS

Reviewer #1 (Remarks to the Author):

The revised manuscript by Choate and colleagues is substantially improved and provides a clear and compelling story. All of my previous comments have been successfully addressed.

Reviewer #3 (Remarks to the Author):

The new version of this manuscript is a fantastic read, and the authors have address all my major concerns. I appreciate their thoughtful replies to my comments, and the care they took with addressing the issues I raised. I look forward to seeing it published.

I have a few minor comments.

I'm not clear what the evidence is for a hard selective sweep. I think the current definition for a hard selective sweep would be a region that has reached fixation in Europeans (it does vary a bit depending who is using it), and the impression I had from reading the paper was that there was still variation in all human populations. It might be helpful for the authors to clarify why they are calling it "hard", and why that is important. (I'm not sure it is - there is copious evidence for a selective sweep, which is exciting whether it is called hard or not).

The description of the 1000G ancestry at line 255: I'm not entirely comfortable with the description of "individuals of European ancestry" and "individuals of African ancestry" considering the likelihood that many are of substantially mixed ancestry (the individuals of African ancestry are almost certainly also individuals of European ancestry). This isn't so much a question of scientific validity - I agree with the authors' interpretation of their results - but of accurately describing individual humans rather than over simplifying their complexity. It would be more accurate to say "individuals confirmed to have some European ancestry" and "individuals confirmed to have some African ancestry" - but this is obviously more wordy and less elegant! I leave it entirely up to the authors as to whether they chose to revise their text here, or if they think it is ok as is - I just wanted to make a note of this potentially sensitive issue.

The final sentence (line 324) would be even more elegant without the second use of the word "continued"

Reviewer #4 (Remarks to the Author):

Note: these comments are coming from a reviewer who did not see the original manuscript and who was asked examine the reviewers' reports and the authors' responses because Reviewer #2 was not available.

Reviewer #2 was enthusiastic about the original version of the manuscript. I agree with this reviewer's positive comments. Her/his concerns involved requests for clarification (points 1 and 3) and reliance on data of convenience (points 1, 2 and 4). These are all fair and constructive points. In my estimation, the authors have responded adequately to each of the concerns.

REVIEWERS' COMMENTS

Reviewer #1 (Remarks to the Author):

The revised manuscript by Choate and colleagues is substantially improved and provides a clear and compelling story. All of my previous comments have been successfully addressed.

Response: We thank the reviewer for all of their helpful comments.

Reviewer #3 (Remarks to the Author):

The new version of this manuscript is a fantastic read, and the authors have address all my major concerns. I appreciate their thoughtful replies to my comments, and the care they took with addressing the issues I raised. I look forward to seeing it published.

Response: We thank the reviewer for their suggestions.

I have a few minor comments.

I'm not clear what the evidence is for a hard selective sweep. I think the current definition for a hard selective sweep would be a region that has reached fixation in Europeans (it does vary a bit depending who is using it), and the impression I had from reading the paper was that there was still variation in all human populations. It might be helpful for the authors to clarify why they are calling it "hard", and why that is important. (I'm not sure it is - there is copious evidence for a selective sweep, which is exciting whether it is called hard or not).

Response: We used the definitions of hard and soft sweeps reported by Messer and Petrov (Trends in Eco Evo, 2013). Under this definition, hard sweeps are sweeps from novel variants or whose ancestral variants were lost (a soft sweep can harden). To avoid confusion, we have removed most of the statements of hard or soft sweeps. We have also weakened the language arguing that the sweep identified using SweepFinder2 is a hard sweep and cite the Messer and Petrov article in our explanation. We agree with the reviewer that these changes were necessary to avoid confusion.

The description of the 1000G ancestry at line 255: I'm not entirely comfortable with the description of "individuals of European ancestry" and "individuals of African ancestry" considering the likelihood that many are of substantially mixed ancestry (the individuals of African ancestry are almost certainly also individuals of European ancestry). This isn't so much a question of scientific validity - I agree with the authors' interpretation of their results - but of accurately describing individual humans rather than over simplifying their complexity. It would be more accurate to say "individuals confirmed to have some European ancestry" and "individuals confirmed to have some African ancestry" - but this is obviously more wordy and less elegant! I leave it entirely up to the authors as to whether they chose to revise their text

here, or if they think it is ok as is - I just wanted to make a note of this potentially sensitive issue.

Response: We agree with the reviewer here. In response, we have decided to split the difference. We now use the language suggested by the reviewer (i.e., “individuals confirmed to have”) when we are introducing this analysis, and use the less clunky and shorter phrasing in the rest of the paragraph.

The final sentence (line 324) would be even more elegant without the second use of the word “continued”

Response: We have removed the second continued from the final sentence.

Reviewer #4 (Remarks to the Author):

Note: these comments are coming from a reviewer who did not see the original manuscript and who was asked examine the reviewers' reports and the authors' responses because Reviewer #2 was not available.

Reviewer #2 was enthusiastic about the original version of the manuscript. I agree with this reviewer's positive comments. Her/his concerns involved requests for clarification (points 1 and 3) and reliance on data of convenience (points 1, 2 and 4). These are all fair and constructive points. In my estimation, the authors have responded adequately to each of the concerns.

Response: We thank the reviewer for their feedback.